# FOXA2 promotes metastatic competence in small cell lung cancer

Kenta Kawasaki[1,2], Sohrab Salehi[3], Yingqian A. Zhan [4], Kevin Chen[5], Jun Ho Lee[1], Eralda Salataj[4], Hong Zhong[2], Parvathy Manoj[2], Dennis Kinyua[2], Barbara P. Mello [2], Harsha Sridhar[2], Sam E. Tischfield [6], Irina Linkov[7], Nicholas Ceglia[3], Matthew Zatzman [3], Eliyahu Havasov[3], Neil J. Shah[2], Fanli Meng[6], Brian Loomis [6], Umesh K. Bhanot [7], Esther Redin[2], Elisa de Stanchina[5], Pierre-Jacques Hamard [4], Richard P. Koche [4], Andrew McPherson [3], Álvaro Quintanal-Villalonga[2], Sohrab P. Shah [3], Joan Massagué [1,8] ✉ & Charles M. Rudin [2,8] ✉

Small cell lung cancer (SCLC) is known for its high metastatic potential, with most patients demonstrating clinically evident metastases in multiple organs at diagnosis. The factors contributing to this exceptional metastatic capacity have not been defined. To bridge this gap, we compare gene expression in SCLC patient samples who never experienced metastasis or relapse throughout their clinical course, versus primary SCLC patient samples from more typical patients who had metastatic disease at diagnosis. This analysis identifies FOXA2 as a transcription factor strongly associated with SCLC metastasis. Subsequent analyses in experimental models demonstrates that FOXA2 induces a fetal neuroendocrine gene expression program and promotes multi-site metastasis. Moreover, we identify ASCL1, a transcription factor known for its initiating role in SCLC tumorigenesis, as a direct binder of the *FOXA2* promoter and regulator of *FOXA2* expression. Taken together, these data define the ASCL1-FOXA2 axis as a critical driver of multiorgan SCLC metastasis.

Small cell lung cancer (SCLC) is a deadly disease characterized by early and often widespread metastasis, and exceptionally poor prognosis. Two thirds of patients have clinically evident distant metastases at diagnosis, and such patients have a median duration of survival of ~1 year[1,2]. Identifying and characterizing key factors responsible for the exceptional metastatic propensity of this disease could define targets to constrain metastasis, ultimately leading to better clinical outcomes for patients with SCLC.

Despite the major clinical need to identify drivers of metastasis, the lack of actionable oncogenic mutations in SCLC that might justify rebiopsy has limited the opportunity to obtain serial histological samples from patients. Very few comparative analyses of human SCLC primary and metastatic tumors have been performed, and to date these have been largely unrevealing. For example, comparison of primary tumors and their matched metastases in 5 autopsy cases with whole-exome sequencing and transcriptome sequencing (RNA-seq)

[1]Cancer Biology and Genetics Program, Sloan Kettering Institute, Memorial Sloan Kettering Cancer Center, New York, NY, USA. [2]Department of Medicine, Memorial Sloan Kettering Cancer Center, New York, NY, USA. [3]Computational Oncology Program, Memorial Sloan Kettering Cancer Center, New York, NY, USA. [4]Center for Epigenetics Research, Memorial Sloan Kettering Cancer Center, New York, NY, USA. [5]Antitumor Assessment Core, Memorial Sloan Kettering Cancer Center, New York, NY, USA. [6]Marie-Josée and Henry R. Kravis Center for Molecular Oncology, Memorial Sloan Kettering Cancer Center, New York, NY, USA. [7]Pathology Core Facility, Memorial Sloan Kettering Cancer Center, New York, NY, USA. [8]Weill Cornell Medicine Graduate School of Medical Sciences, New York, NY, USA. ✉e-mail: massaguj@mskcc.org; rudinc@mskcc.org

failed to identify candidate drivers of metastasis[3]. In addition to the small sample size, inter-patient heterogeneity, and the fact that primary tumors in such cases may already have the necessary framework for promoting metastasis, all contribute to the lack of knowledge on metastatic drivers in SCLC.

Genetically engineered mouse models (GEMM) of SCLC have proven utility for studying various aspects of this disease but lack some key characteristics of human SCLC including a high incidence of metastases to the brain: SCLC GEMMs metastasis is mostly limited to lymph nodes and liver[4]. Analyses of multiple SCLC GEMMs have suggested that intratumoral heterogeneity can enable metastatic competency[5,6]. Nuclear factor I B, a transcription factor encoded by the *Nfib* gene, is frequently upregulated in the context of metastasis in murine SCLC[7,8]. While NFIB promotes tumor progression in murine SCLC, deletion of *NFIB* did not prevent metastasis, and its role in human disease remains undefined[9,10]. Recent studies in SCLC GEMM with and without *Nfib* knockout suggested an association of FOXA1 and/or FOXA2 transcription factors with metastasis[10], but experimental validation of these candidates has not been reported.

The FOX (forkhead box) family of transcription factors are critical regulators of multiple aspects of early mammalian development and organogenesis[11]. Foxa2 is one of the first members of this family to be expressed during murine embryogenesis. Foxa2 is expressed throughout the endoderm as it gives rise to the lung and to other visceral organs[12,13]. The FOX family members are considered to be "pioneer factors," determining sequential developmental progression through opening of chromatin to facilitate the access of other tissue-specific transcription factors to promoter and enhancer elements throughout the genome[14].

In contrast to the paucity of data on metastatic drivers of human SCLC, there have been detailed analyses of determinants of metastasis for lung adenocarcinoma (LUAD), breast cancer, and other solid tumors in both human and murine systems. Recent analyses of LUAD suggest that genetic mutations are driving oncogenesis, while metastasis is driven primarily by non-genetic mechanisms[15]. LUAD metastatic colonization is enriched for a continuum of developmental stem-like progenitor cell states, from primitive *SOX2*+ progenitors through later stage *SOX9*+ progenitors[16]. The concept of non-genetically defined metastasis-initiating cell states is similarly implicated in colon adenocarcinoma[17–19] and may apply more broadly across solid tumors[20].

Studies of breast and LUAD metastasis have identified determinants of differential organ tropism, including tumor-intrinsic and -extrinsic factors[21–24]. SCLC characteristically metastasizes to multiple sites, frequently including the contralateral lung, brain, liver, adrenal glands, and bone[1]. Whether SCLC tumors differ in organ tropism, or whether factors contributing to the intrinsically high metastatic predilection of this cancer type similarly promote spread to all sites, has not been determined.

In this study, we sought to identify drivers of SCLC metastasis through analysis of SCLC patient cohorts, with subsequent testing and validation in experimental models. These analyses identify FOXA2 as a determinant of SCLC metastasis. FOXA2 in SCLC contributes to the generation of a fetal neuroendocrine gene expression program and promotes multi-site metastasis in vivo. ASCL1, a transcription factor known to serve a critical role in SCLC tumorigenesis, binds to the *FOXA2* promoter and is necessary to drive *FOXA2* expression in SCLC. Taken together, these data define the ASCL1-FOXA2 axis as a driver of SCLC metastasis.

## Results

### Most SCLC demonstrate early multi-organ metastatic capacity

It is well established that most patients with SCLC have extensive-stage disease, with radiologically evident metastases at the time of diagnosis. To characterize with more granularity the distribution of disease at initial presentation, we profiled a cohort of 327 patients diagnosed with SCLC, and selected 268 patients from the cohort all of whom underwent a complete staging workup including $^{18}$F-flourodeoxyglucose positron emission tomography/computed tomography ($^{18}$F-FDG PET/CT) and brain magnetic resonance imaging (MRI) or computed tomography (CT) within 3 months of initial visit (Fig. 1a). Consistent with prior reports[1], 61% of patients had radiologically overt distant metastases at the initial visit, with the most frequent sites of metastatic spread being distant lymph nodes, lung (contralateral or different lobe), bone, and liver (Fig. 1a). Of the 39% of patients without distant metastases, 15% were notable in that they had a single isolated lung mass at diagnosis, no nodal involvement, underwent surgical resection or concomitant chemoradiation with curative intent, and had no evidence of subsequent disease recurrence (median follow-up, 4.6 years). The existence of such patients is consistent with at least two alternative hypotheses: (1) these cases were "caught early" but are biologically indistinguishable from advanced disease, or (2) these cases were intrinsically different, lacking one or more factors required for metastasis. We refer to these as "never metastatic" ("never-met", for short) SCLC, in contrast to "metastasis-associated" ("met-associated") primary SCLCs.

Of the 61% of patients with radiologically evident distant metastases at their initial visit, over 90% had multiple concurrent metastases (Fig. 1a). A wide distribution of metastatic sites was observed (Supplementary Fig. 1a). In the patients with only one radiologically evident metastatic site at diagnosis for whom at least 12 months of clinical follow-up was available, new sites of metastasis emerged in 8 out of 9 patients (Supplementary Fig. 1b). These observations suggest that SCLC typically contains metastasis progenitor cells without strongly selective organotropism – i.e. with diffuse intrinsic metastatic capacity and high adaptability to the organs they seed. Alternatively, these tumors might contain a diversity of distinct metastasis-initiating subclones with differential predilection for site-specific metastatic colonization.

To explore these alternative hypotheses, we adopted approaches previously applied successfully in the context of breast adenocarcinomas to isolate cancer cell subpopulations with differential organotropism through selective harvesting and systemic reintroduction of organ-specific metastases[21,22,25]. Accordingly, we performed serial selection in immunodeficient NSG (NOD.Cg-*Prkdc^scid Il2rg^{tm1Wjl}*/SzJ) mice using two SCLC cell lines (H1836 and H82) and a patient-derived xenograft model (PDX Lx773I) (Fig. 1b, Supplementary Fig. 1c). SCLC has been classified based on differential expression of lineage defining transcription factors ASCL1, NeuroD1, and POU2F3, and we chose one model representative of each of these three subtypes (Supplementary Fig. 1d). Single-cell suspensions were inoculated into the left cardiac ventricle to enable arterial dissemination of the cells to multiple organs[26]. Metastatic tumors were harvested after 1–3 months and cancer cell populations were isolated from multiple metastatic sites including liver, adrenal, brain, and ovary, and were expanded in culture.

These cell sublines were inoculated into new sets of NSG mice by intra-cardiac injection. While these SCLC sublines were derived from different organs, they demonstrated consistently similar organ distribution patterns in subsequent rounds of metastasis upon reinoculation, with no evident enrichment for tropism to the organ from which they were isolated (Fig. 1c). Thus, we found no evidence of enriched or selective organotropism in SCLC subclones that were subjected to a limited number of rounds of in vivo selection. These findings are in contrast with the marked organotropism exhibited by triple-negative breast cancer cell lines that were previously subjected to the same protocol[21,22,25]. Taken together, these data suggest that individual organ-specific SCLC metastases retain a broad metastatic competency consistent with that of the tumor from which they were derived.

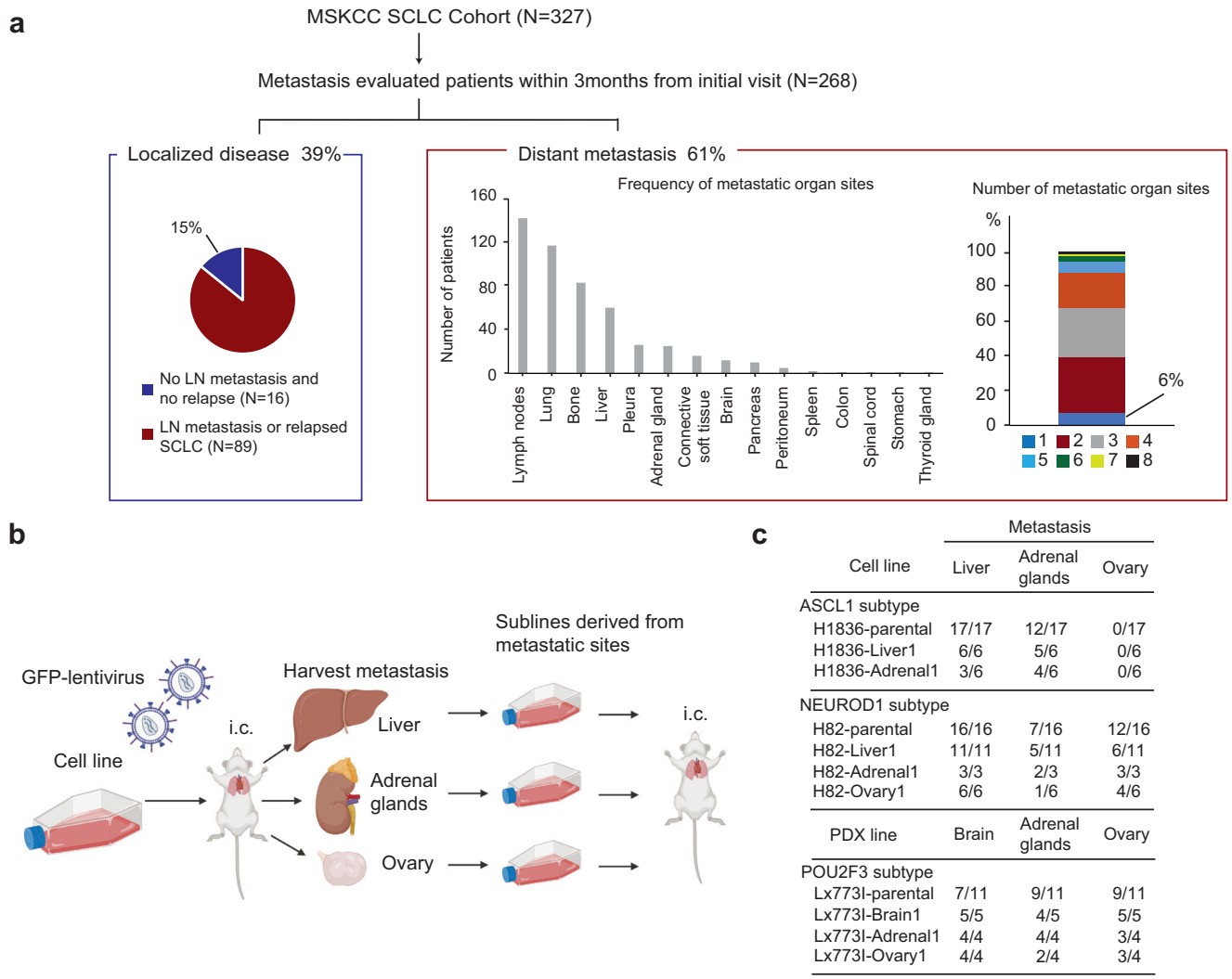

**Fig. 1 | Metastatic characteristics of SCLC. a** Metastatic characteristics of an MSK-SCLC cohort (*N* = 327). Localized disease (left) including the fraction of never-metastasized cases (dark blue). Frequency (middle) and number (right) of distant metastatic sites are shown among cases with any metastases. LN: Lymph node. **b** Schematic of the approach to metastatic subline generation in SCLC cell lines. i.c.: intra-cardiac injection. Created in BioRender. Rudin C. (2025) https://BioRender.com/j5rkxx5. **c** Frequency and distribution of metastases in H1836 (ASCL1 subtype) and H82 (NEUROD1 subtype) SCLC cell lines, and Lx773I SCLC PDX (POU2F3 subtype) parental lines and organ-specific sublines after intracardiac implantation.

## Xenotransplantation as a surrogate for metastatic competence

Our laboratory has established an extensive library of PDX models, including from patients with SCLC[27]. We identified 41 cases in which we attempted to xenograft SCLC from biopsies of the primary lung tumor. Among 33 biopsies of patients who presented with or developed metastatic disease, 52% yielded PDX. In contrast, among 8 biopsies of patients whose disease never metastasized (no lymph node metastasis and no recurrence with median follow-up of 3.2 years), none yielded a PDX (Supplementary Fig. 1e). To the extent that successful xeno-transplantation into the mouse flank may reflect a capacity for tumor regeneration essential for metastatic spread, these observational data support the hypothesis that there are intrinsic differences in meta-static competency between never-met and met-associated primary SCLCs.

## Identification of candidate drivers of SCLC metastasis

To identify potential determinants of metastatic capacity in SCLC, we explored the hypothesis that the rare never-met SCLCs might repre-sent tumors that lack essential drivers of metastasis. We identified 8 SCLC cases meeting strict criteria as never-metastatic, including pathologic stage T1-3N0M0 (no evidence of spread, even to local

intralobar nodes), definitive treatment with surgical resection or con-comitant chemoradiation, no relapse with a minimum of 2 years of documented follow-up, and available tumor material (age:59−75, Male 25%). This 2 year milestone was chosen given that 90-95% of SCLC cases recur and progress within 1−2 years. As a comparator, we chose 54 patients (age:27-88, Male 61%) who presented with distant meta-static SCLC at diagnosis, including 5 from whom diagnostic biopsy material for both DNA and RNA analysis was available from the primary lung tumor (cases we refer to as met-associated primary SCLC) (Fig. 2a, and Supplementary Data 1). The tumor sizes of the early-stage treated cases without recurrence varied from 1.5 to 4.5 cm in maximal dia-meter (T1-3) (Supplementary Fig. 2a). There was no significant differ-ence in Ki-67 index between never-met and met-associated tumors (Supplementary Fig. 2b). Over 80% of the patients with metastatic disease at diagnosis died within 2 years; in contrast, none of the never-met SCLC cohort experienced disease recurrence or death from any cause within the period of minimal follow-up (Supplementary Fig. 2c, d).

We initially assessed genomic profiles of never-met and met-associated SCLC cases by MSK-IMPACT, a clinical targeted sequencing platform covering over 500 cancer-related genes (Fig. 2b)[28,29]. *TP53*

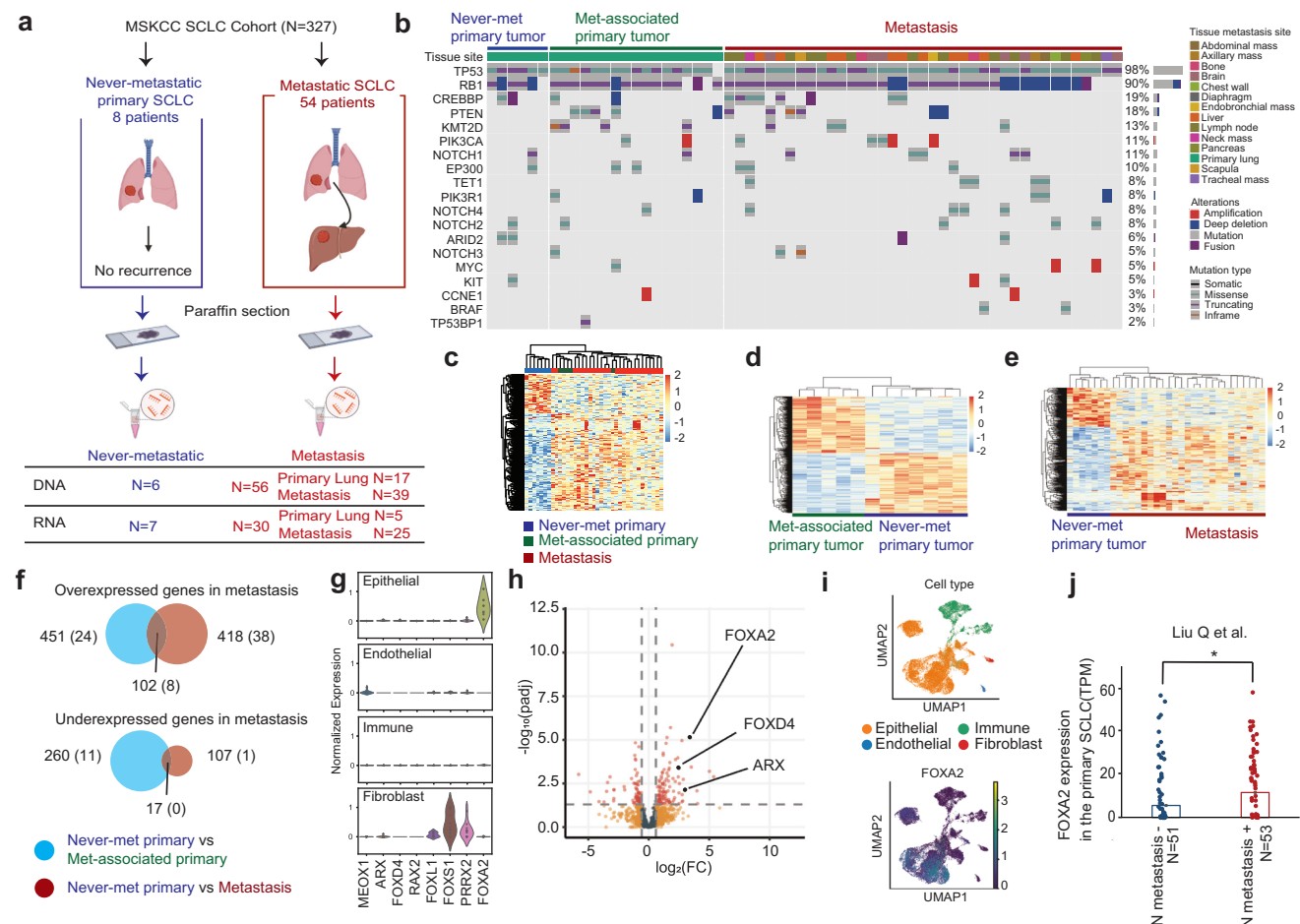

**Fig. 2 | Genetic and expression profiles of the SCLC cohort. a** Sample selection schema of never-metastatic (left) and metastatic SCLC. Created in BioRender. Rudin C. (2025) https://BioRender.com/qkiri1o. **b** Prevalent mutations in never-metastasized primary SCLC and metastatic SCLC cohort. **c** Heatmap of never-met primary SCLC, metastasis-associated (met-associated) primary SCLC, and metastatic SCLC. **d** Heatmap of never-met primary SCLC and met-associated primary SCLC. **e** Heatmap of never-met primary SCLC and metastatic SCLC. **f** Overlapping factors from panel (**d**, **e**). The data are analyzed with Wald's test and 8 transcription factors were nominated following the criteria of FC > 4 or FC < −4, $P_{adj} < 0.05$. The number of transcription factors is indicated in brackets. **g** Violin plot of expression for each of the 8 candidates based on scRNA-seq data of primary SCLC. **h** Volcano plot of candidate genes in bulk RNA-seq data comparing the expression of met-associated primary SCLC to never-met primary SCLC. **i** UMAP of FOXA2 in each component from scRNA-seq. **j** FOXA2 expression in primary SCLC with or without lymph node (LN) metastasis from the data of Liu Q et al. Graph is shown as mean ± SEM (* < 0.05, *P* = 0.04, two-sided Student's *t*-test).

and *RB1* mutations were the most frequently seen mutations in both cohorts, in line with the highly frequent presence of these mutations in SCLC[30]. No consistent mutational differences between cohorts were observed, including tumor mutation burden (TMB) (Supplementary Fig. 2e). This result was consistent with broader sequencing analyses suggesting that non-genetic mechanisms, rather than specific somatic mutations, are primary determinants of metastatic progression[15].

We hypothesized that a unique transcriptional state tied to the differential expression of transcription factors determines metastatic competence in SCLC. Sufficient material for expression profiling by RNA-seq analysis was available from 7 never-met SCLC tumors, and from 30 metastatic tumors including 5 with their met-associated primary tumor material available. We hypothesized that primary SCLC tumors that never metastasized might lack one or more factors involved in metastatic competency, which would be present in primary SCLC tumors that gave rise to metastatic disease, and which would presumably also be retained in metastatic SCLC. In addition to comparing gene expression in never-met primary tumors vs. met-associated primary tumors and metastases considered jointly (Fig. 2c). To minimize the impact of distinct microenvironments on the transcriptional profiles, we initially analyzed differential gene expression between the never-met primary SCLC tumors versus met-

associated primary SCLC tumors (Fig. 2d, Supplementary Fig. 2f). Comparison of these two cohorts by unsupervised clustering revealed different gene expression profiles, consistent with the hypothesis that these cohorts are biologically distinct (Fig. 2d, Supplementary Fig. 2f).

We also compared expression profiles of the 7 never-met cases with 25 samples from metastatic sites. Despite greater expression heterogeneity in this cohort, there were still evident differences between never-met primary tumors and metastatic samples (Fig. 2e, Supplementary Fig. 2g). We hypothesized that the most promising candidate drivers are those showing consistent differences between the two cohorts of primary tumors that are retained as differences between the never-met tumors and the metastases. The overlapping common genes between these two comparisons included 102 overexpressed and 17 underexpressed genes associated with metastasis (Fig. 2f and Supplementary Data 2) (FC > 4 or FC < −4, $P_{adj} < 0.05$).

Since distinct transcriptional programs might underlie differential metastatic capacities or different progenitor cell states, we focused on transcription factors[31]. There were no transcription factors among the genes significantly underexpressed in metastatic tumors, while 8 transcription factors were among the overexpressed genes (Fig. 2f). Within the metastatic cohort, expression of these 8 transcription factors did not differ between met-associated primary lung tumors (N = 5)

and metastases ($N = 25$), while they were significantly underexpressed in never-met tumors ($N = 7$) (Supplementary Fig. 2h). This was consistent with the possibility that primary tumors acquired the necessary transcriptional framework for metastatic competency. We obtained a similar result by querying a publicly available RNA-seq dataset from 5 SCLC autopsy cases with matched biopsies of primary met-associated and metastatic lesions (Supplementary Fig. 2i)[3]. Notably the expression level of *NFIB*, a previously proposed metastasis gene[7,10], did not differ between the never-met and met-associated cohorts (Supplementary Fig. 2j).

This approach identified 8 transcription factors as candidates in promoting SCLC metastasis. Differential expression of these factors in bulk RNA-seq could be derived from the cancer cells, or alternatively from other cells within the tumor microenvironment. To distinguish between these possibilities, we turned to our previously published human SCLC single cell RNA-seq (scRNA-seq) dataset[32]. A heatmap of differential gene expression for the 8 candidates demonstrated *FOXA2* as the top candidate uniquely expressed in the epithelial (SCLC) cells (Fig. 2g). In contrast, most other candidates were predominantly expressed in endothelial, fibroblast, or immune cell clusters. Among the candidates expressed in cancer cells, differential expression between never-met versus met-associated SCLC primary tumors was greatest for *FOXA2* (Fig. 2h, Supplementary Fig. 2k). A UMAP (Uniform Manifold Approximation and Projection) embedding of scRNA-seq data from 11 primary SCLC lung tumors confirmed that *FOXA2* expression was primarily restricted to the cancer cell compartment in SCLC tumors (Fig. 2i).

To further explore the association of our lead candidates with SCLC metastatic capacity, we interrogated a second, fully independent, set of 41 SCLC using the same criteria to classify the tumors as never-met ($N = 3$) and met-associated ($N = 38$) SCLC (Supplementary Fig. 2l). While this cohort was statistically underpowered, *FOXA2* again emerged as a differentially expressed gene in met-associated SCLC tumors (Supplementary Fig. 2m,n).

Finally, we took advantage of a recently published transcriptomic dataset from resected primary SCLC in a cohort of patients with ($N = 53$) and without ($N = 51$) intrathoracic lymph node metastasis[33]. Further supporting this association, primary tumors with lymph node metastases had significantly higher expression of *FOXA2* than primary tumors without (Fig. 2j).

## Prognostic significance of FOXA2 in limited stage SCLC
To further interrogate the association of FOXA2 with the clinical outcome of SCLC patients, we prepared a tissue microarray (TMA) containing tumors from 26 cases of limited stage SCLC treated by surgical resection followed by adjuvant chemotherapy (age:55-79, Male 58%, Supplementary Data 3). These cases were fully independent from the clinical cohorts analyzed above. Among these 26 patients, 15 experienced metastatic recurrence during follow-up. FOXA2 protein expression was assessed by immunohistochemistry (IHC) and an H-score assigned by a pathologist blinded to sample identity. A comparison of H-scores showed a significantly lower expression of FOXA2 in tumors that never recurred, with all but two tumors without recurrence demonstrating an H-score of zero (Fig. 3a). Based on the H-score, we explored an optimal discriminant value for dichotomizing FOXA2^high vs. FOXA2^low by ROC (receiver operating characteristics) analysis (Supplementary Fig. 3a). Using this classifier, 4 out of 13 patients with FOXA2^low tumors experienced metastatic recurrence, versus 11 out of 13 patients with FOXA2^high expression (Supplementary Fig. 3b). Based on relapse-free survival in these patient cohorts, patients with FOXA2^high SCLC had significantly worse prognosis compared to patients with FOXA2^low SCLC (Fig. 3b). Together these data identify FOXA2 as a potential predictive biomarker of disease recurrence in patients with definitively treated limited stage SCLC.

## FOXA2 promotes SCLC metastasis in vivo
With the totality of clinical data consistently pointing to FOXA2 as a promising candidate, we focused on further exploring the potential role of FOXA2 as a promoter of metastatic competence in SCLC experimental model systems. To this end, we evaluated the effects of FOXA2 manipulation in SCLC cell lines. To assess the effect of *FOXA2* knock-down (KD), we selected two primary lung-derived SCLC cell lines that expressed FOXA2 (H1836 and SHP-77). The cell lines were stably transduced with a lentiviral vector with either of two doxycycline-inducible *FOXA2* shRNAs or a safe-targeting control shRNA. Specific suppression of FOXA2 protein expression was confirmed by western immunoblotting (Supplementary Fig. 3c). The cells were also marked with luciferase to allow longitudinal quantitative analysis in vivo by bioluminescence imaging (BLI). Samples containing $5 \times 10^5$ viable cells of each transduced line were introduced by intra-cardiac injection into 6–8 week old NSG mice. Mice were given doxycycline-containing chow (2 g/kg) starting 1 week prior to SCLC cell injection. Following intracardiac injection, BLI signals were analyzed weekly. BLI signals generated from the KD groups showed significant decrease compared to the control group in both cell lines (Fig. 3c, e). The burden of macroscopic metastases in mice euthanized on day 49 (SHP-77) or day 56 (H1836) was consistent with the results of BLI, demonstrating a markedly lower number of liver metastases, the most frequent site of metastasis, with *FOXA2* KD (Fig. 3d, f). Histological analysis confirmed reductions in the number of metastatic lesions in all *FOXA2* KD cohorts relative to controls (Supplementary Fig. 3d, e). Attempts to establish FOXA2 knockdown in a 3rd cell line (H2081, H146, one PDX derived line) did not yield a sufficient reduction in FOXA2 expression to move to in vivo experiments.

The predominance of liver metastases from these FOXA2-high cell lines raised the question of whether FOXA2, in addition to promoting metastasis, might alter organ tropism toward liver metastasis. We further explored the distribution of FOXA2 levels among 25 metastatic sites in our patient cohort. Liver metastases do not appear to be enriched among high FOXA2 tumors relative to other metastatic sites (Supplementary Fig. 3f).

To further demonstrate specificity of metastatic inhibition by FOXA2 KD, we performed an add-back experiment in the SHP-77 KD line. Human and murine *FOXA2* demonstrate 97% homology[34]. Despite this close homology, one of the *FOXA2* shRNA used was human specific, allowing us to restore FOXA2 activity in KD cells with murine *FOXA2* cDNA. Restoration of FOXA2 expression was confirmed by western blot (Supplementary Fig. 3g). Intra-cardiac injection of FOXA2-restored cells in mice showed clear re-establishment of metastatic potential in *FOXA2* KD cells (Fig. 3g, h), confirming that FOXA2 is indeed required in this line for high metastatic competency.

Next we sought to investigate the effects of exogenous FOXA2 expression in a FOXA2-low cell line, H1963. Intracardiac injection of H1963 resulted in rare metastases, exclusively to the ovary (1 out of 6, and 1 out of 8 mice, in two independent experiments). The ovarian metastases demonstrated upregulation of FOXA2 by immunohistochemistry relative to the subcutaneous tumor generated from the parental line (Fig. 3i), implying that subclones expressing FOXA2 might be selectively contributing to metastasis. An H1963 subline generated from an ovarian metastasis (Fig. 3j) indeed showed persistent strong enrichment of FOXA2 relative to the parental line (Fig. 3k). Thus FOXA2 is strongly associated with metastasis and appears to be required for high metastatic competency in FOXA2-high SCLC.

## FOXA2 promotes a fetal neuroendocrine gene expression program
While the observed decrease in metastatic activity caused by *FOXA2* KD was consistent with a role of FOXA2 in metastasis, an alternative explanation might be that *FOXA2* KD has a general suppressive effect on SCLC cell proliferation and tumor growth. To distinguish between

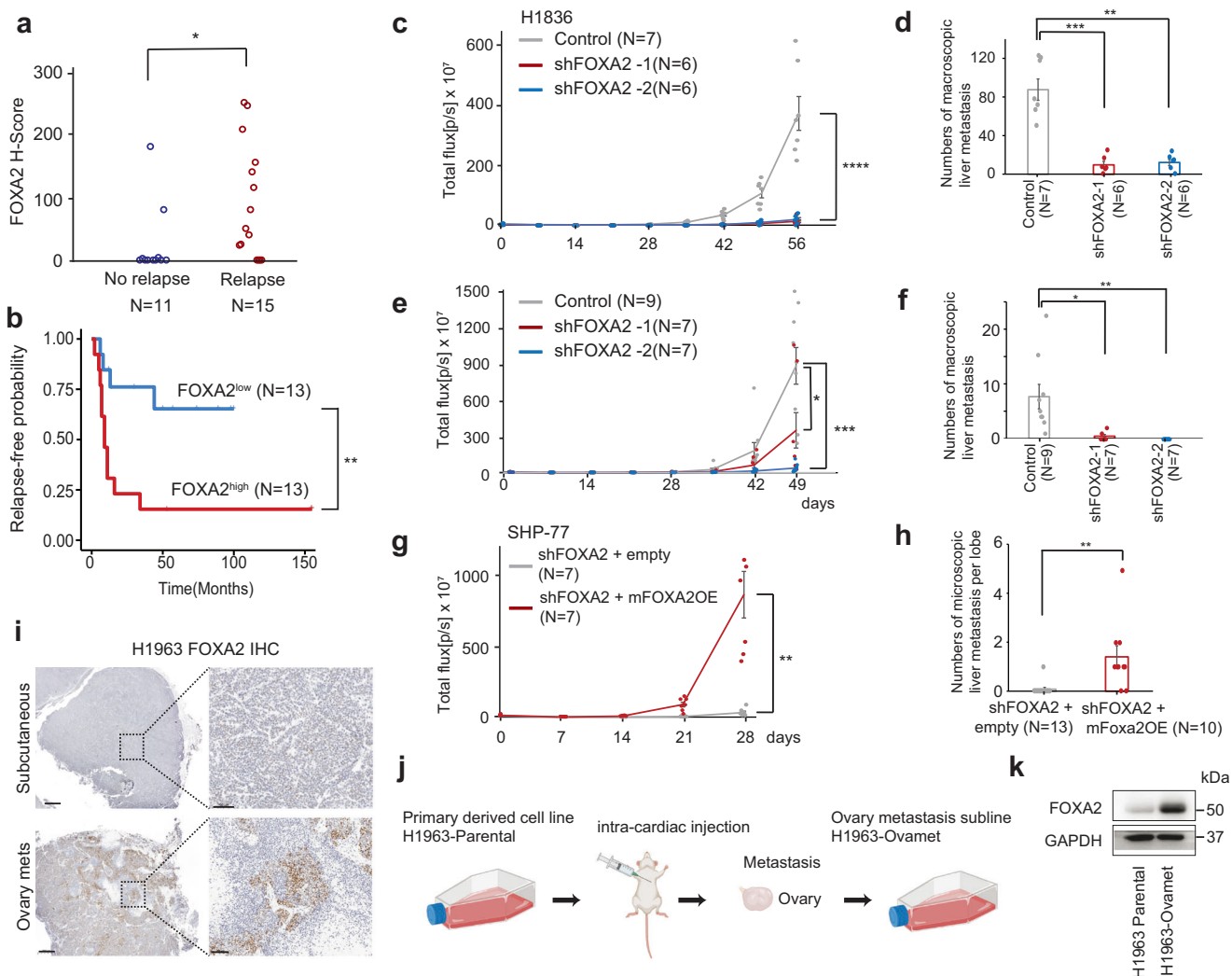

**Fig. 3 | FOXA2 promotes SCLC metastasis. a** Clinical validation using the TMA tissue sections. FOXA2 H-score of a TMA of limited stage SCLC patients comparing those with no relapse ($N = 11$) vs. relapse ($N = 15$) (* < 0.05, $P = 0.049$, two-sided Student's *t*-test). **b** Relapse-free survival of patients with FOXA2^low and FOXA2^high tumors (** < 0.01, $P = 0.0033$, log-rank test). **c–f** FOXA2 KD with short-harpin RNA (sh) in cell lines H1836 (**c**, **d**) and SHP-77 (**e**, **f**) and metastatic burden assessment. Total photon reflux [p/s] is indicated in graphs (**c**, **e**), and the number of macroscopic metastases after euthanasia are indicated in graphs (**d**, **f**). Mean ± SEM (* < 0.05, ** <0.01, *** <0.001, **** <0.0001, one-way ANOVA. **c**; $P = 6.0 \times 10^{-4}$(top), $P = 7.0 \times 10^{-4}$(bottom), **d** $P = 4.6 \times 10^{-4}$(left), $P = 1.0 \times 10^{-3}$(right), **e** $P = 0.033$(top), $P = 2.6 \times 10^{-4}$(bottom), **f** $P = 0.014$(left), $P = 9.6 \times 10^{-3}$(right)). **g**, **h** Mouse Foxa2 overexpression (OE) in KD cell line SHP-77. Total photon reflux [p/s] is indicated in graph (**g**) and the number of microscopic liver metastases in each liver lobe after euthanasia are indicated in graph (**h**). Mean ± SEM (** < 0.01, **g**; $P = 2.3 \times 10^{-3}$, **h**; $P = 3.6 \times 10^{-3}$, two-sided Student's *t*-test). **i** FOXA2 immunohistochemistry (IHC) of H1963 subcutaneous tumor and its ovarian metastasis generated by intra-cardiac injection. Scale bar: 500 μm and 100 μm (inset). **j**, **k** Schematic of the generation of ovarian metastasis (Ovamet) subline (**j**) and FOXA2 western blot of parental and ovarian metastasis subline (**k**). Western blot result is a representative data of one of the 3 individual experiments. Created in BioRender. Rudin C. (2025) https://BioRender.com/58ppd5o.

these alternatives, we analyzed the effect of FOXA2 downregulation in SCLC cell lines using the inducible shRNA models. *FOXA2* KD had no effect on the proliferation rate of H1836 or SHP-77 cells in culture (Supplementary Fig. 4a). We also evaluated the effect of *FOXA2* KD on the growth of these cells as subcutaneous tumors in mice. Tumors were evaluated by weekly measurement (Fig. 4a, d), BLI (Fig. 4b, e), and direct measurement after tumor collection (Supplementary Fig. 4b). Consistently across these assays, *FOXA2* KD did not suppress subcutaneous tumor growth. In one of the two sh*FOXA2* H1836 cell clones, subcutaneous growth appeared to be enhanced, while in all other instances, the tumorigenic growth of *FOXA2* KD cell clones was similar to control. The proliferating cell index as determined by Ki67 immunostaining was not significantly different between control and *FOXA2* KD subcutaneous tumors (Fig. 4c, f). Together these results suggest

that *FOXA2* KD suppresses metastasis of SCLC cells without inhibiting the intrinsic proliferative capacity of these cells.

To begin to explore the basis for the pro-metastatic effect of FOXA2 in SCLC, we performed pathway analyses on global gene expression changes between *FOXA2* KD and safe-targeting controls in the two SCLC cell lines, H1836 and SHP-77. FOXA2 expression was associated with signatures of various embryonic development pathways, notably including a "fetal lung neuroendocrine" cell signature in both cell lines (Fig. 4g, h and Supplementary Data 4). To further probe the associated pathways, we also performed gene signature analysis in H1963, a SCLC line with low FOXA2 expression, versus an H1963 derivative expressing exogenous FOXA2 (Fig. 4i and Supplementary Data 4). Fetal lung neuroendocrine cells signature again appeared as a top pathway significantly associated with FOXA2. The effects of *FOXA2*

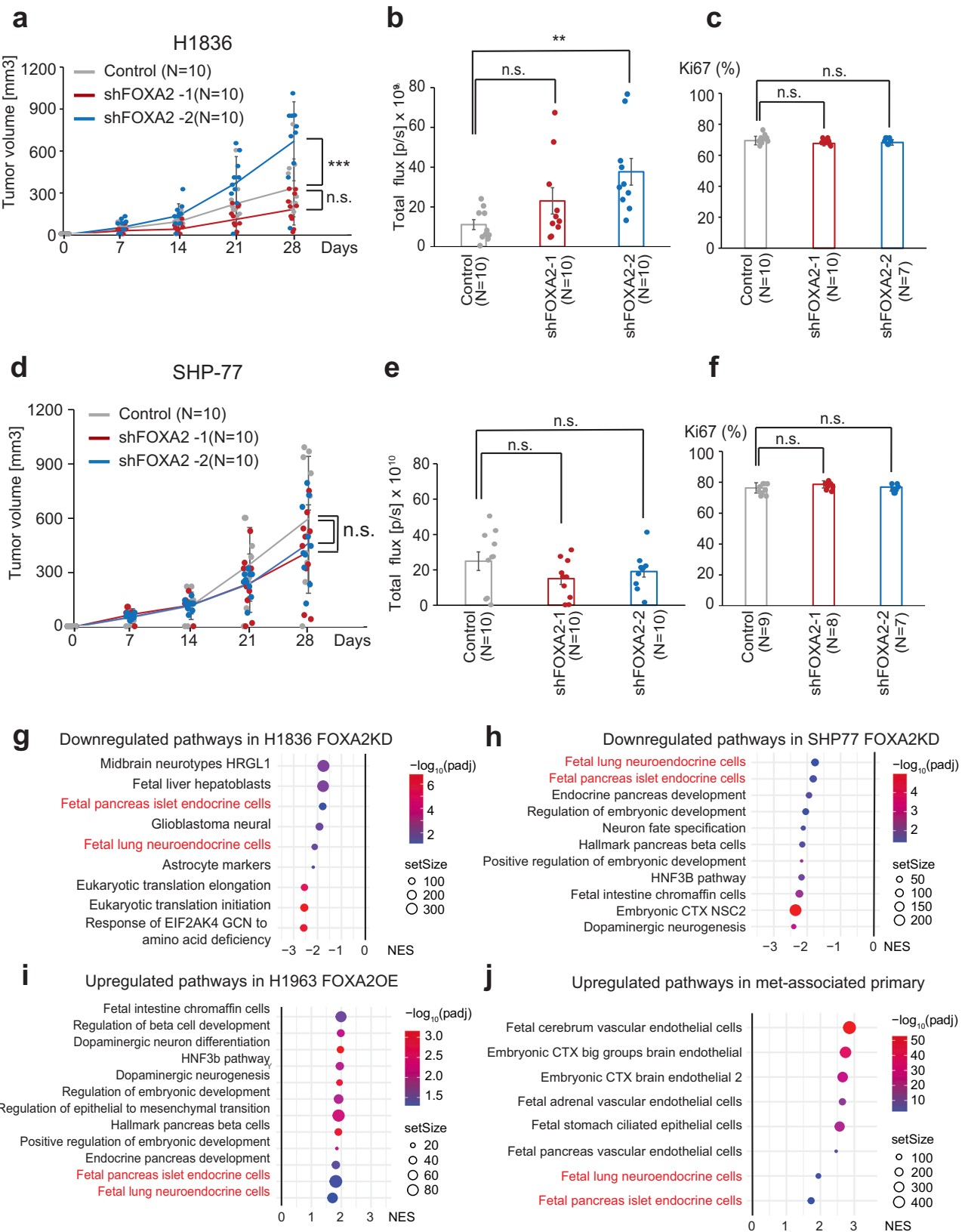

KD and FOXA2 overexpression (OE) was confirmed in the three cell lines as shown in the volcano plot of differential gene expression (Supplementary Fig. 4c). In addition to the "fetal lung neuroendocrine" cell signature, a second endocrine progenitor signature, "fetal pancreas islet endocrine cells", was also consistently associated with FOXA2 expression (Fig. 4g-i). In line with the in vivo data, colony formation assays showed that *FOXA2* KD in both K1836 and SHP-77 resulted in fewer colonies and *FOXA2* OE in H1963 leads to increased number of colonies in vitro (Supplementary Fig. 4d).

In our original clinical dataset, RNA-seq analysis of never-met versus met-associated primary SCLC clinical cohorts also demonstrated an enrichment for several embryonic and fetal signatures

**Fig. 4 | FOXA2 effects on proliferation and gene expression. a–f** Subcutaneous tumor growth of FOXA2 KD and control cell lines (H1836, SHP-77). Tumor proliferation and growth as measured by tumor volume (**a, d**), BLI (**b, e**), and Ki67 score (**c, f**). Graphs indicate mean±S.D. (*** < 0.001, one-way ANOVA. **a** $P = 4.2 \times 10^{-4}$, **b** $P = 0.012$). **g–i** Pathway analysis of RNA-seq data on FOXA2 KD or OE cell lines compared to FOXA2 wild control. This GSEA analysis was based on the 3 replicates submitted for RNA-seq for each cell line. Top pathways downregulated in FOXA2

KD cell lines (H1836 (**g**), SHP-77 (**h**)) and upregulated in the FOXA2 OE cell line (H1963 (**i**)) are indicated."Set size" refers to the total number of genes in the pathway and the exact number and genes can be traced in the supplementary data. **j** Upregulated pathways in met-associated primary tumor compared to never-met primary tumor. This analysis is generated from 1 RNA-seq data set (the same set from Fig. 2) of clinical samples using GSEA pathway analysis. N.s. Not significant, NES normalized enrichment score.

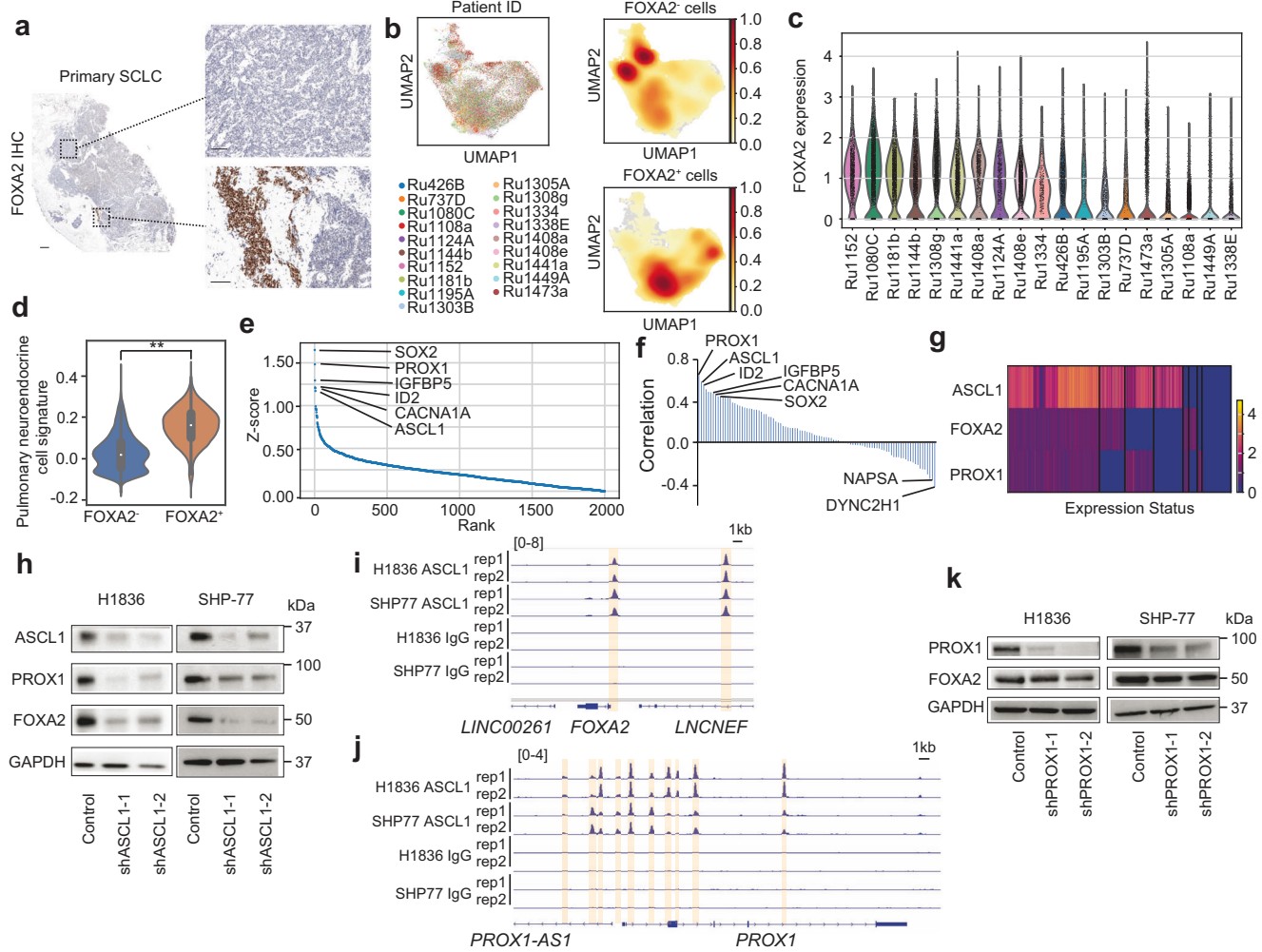

**Fig. 5 | Gene associations and regulation of FOXA2 expression. a** FOXA2 IHC staining of primary SCLC. Representative images of a FOXA2-negative region and a FOXA2-positive region are shown. Scale bar: 500 μm and 100 μm (inset). **b** scRNA-seq UMAP including 19 clinical metastatic SCLC tumors indicating FOXA2⁺ and FOXA2⁻ cells. **c** Violin plot of FOXA2 expression by scRNA-seq in each of these 19 cases. **d** Pulmonary neuroendocrine cell signature expression in FOXA2⁺ cells and FOXA2⁻ cells assessed by scRNA-seq (** < 0.01, $P = 0.0000$, one-sided Mann–Whitney $U$ test). **e** Ranking plot of genes highly expressed in FOXA2⁺ vs. FOXA2⁻ cells at the single-cell level. **f** Correlation plot of nominated top 100 genes from panel (**e**) assessed relative to FOXA2 expression in bulk RNA-seq data. **g** Heatmap of co-expression on ASCL1, FOXA2, and PROX1 in scRNA-seq. **h** ASCL1

KD effect on protein expression of PROX1, FOXA2, and GAPDH (control) in H1836 and SHP-77 cell lines. The samples derive from the same experiment but different gels for ASCL1, FOXA2, another for PROX1 and GAPDH were processed in parallel. Western blot result is a representative data of one of the 3 individual experiments. **i, j** ASCL1 ChIP-seq at the *FOXA2* and *PROX1* gene loci in 2 SCLC cell lines (H1836 and SHP-77). Two independent replicates using anti-ASCL1, and two isotype (IgG) controls are shown. Highlighted regions indicate promoter and enhancer domains. **k** *PROX1* KD and protein expression of FOXA2 and GAPDH (control) in H1836 and SHP-77 cell lines. The samples derive from the same experiment but different gels for FOXA2, another for PROX1 and GAPDH were processed in parallel. Western blot result is a representative data of one of the 3 individual experiments.

associated with metastasis (Fig. 4j). Only two pathways were shared as significant associations with FOXA2 in all three cell line analyses and in the clinical samples: the two fetal pathways, with "fetal lung neuroendocrine cells" having the stronger normalized enrichment score in the clinical data. Taken together these data suggest that the association between FOXA2 expression and metastatic capacity in SCLC might

involve promotion of a progenitor cell state through activation of fetal lung neuroendocrine pathways.

### Characterization of the FOXA2⁺ cell state
While some SCLC metastases demonstrated strong and consistent FOXA2 expression by IHC, others demonstrate both FOXA2-positive

and -negative cancer cells in the same tumor section (Fig. 5a, Supplementary Fig. 5a). To better resolve the transcriptional programs associated with FOXA2, we analyzed single cell (sc) RNA-seq data from 19 metastatic tumors with FOXA2 expression in at least 200 tumor cells in each (age:39-73, Male 42%, Fig. 5b, c, Supplementary Fig. 5b, and Supplementary Data 5). We assessed differential gene expression between FOXA2$^+$ ($N = 19,617$) and FOXA2$^-$ cancer cells ($N = 16,643$) within these SCLC tumors. We used signatures from a human fetal lung single-cell atlas[35] to assess fetal pathway activation in FOXA2$^+$ cancer cells. Consistent with the bulk RNA expression data above, we observed an increased expression of defined lung progenitor single cell signatures such as "fetal lung pulmonary neuroendocrine precursor" in FOXA2$^+$ compared to FOXA2$^-$ cancer cells (Fig. 5d).

Analysis of differential gene expression revealed several genes of interest strongly associated with *FOXA2* in the single cell dataset, including *SOX2*, *PROX1*, and *ASCL1*, all of which encode known regulators of embryonic neuroendocrine development (Fig. 5e, and Supplementary Data 6). To pursue factors that might be regulating FOXA2 from these candidates, we focused on genes with a high correlation in the previous bulk RNA-seq ($N = 37$). We cross-referenced the top 100 genes identified by single cell profiling with this bulk RNA-seq dataset and calculated the correlation of each gene with *FOXA2* (Fig. 5f, Supplementary Fig. 5c). This identified *PROX1* and *ASCL1* as the highest ranked genes. A heatmap from scRNA-seq confirmed that these 3 genes are frequently co-expressed (Fig. 5g).

ASCL1 is a master regulator of neuroendocrine development[36] and is the primary determinant of the largest transcriptionally defined subtype of SCLC[37]. *Ascl1* has been found to be required for tumorigenesis in a SCLC GEMM[38]. Prior analyses suggest that ASCL1 regulates *Foxa2* expression in mouse models and cell lines[38,39]. PROX1 is a transcription factor expressed in embryogenesis. *PROX1* has also been reported as a downstream target of ASCL1[40].

To interrogate the interdependence of expression of *PROX1* and *FOXA2*, we assessed the effects of KD of *ASCL1* and of *PROX1* in SCLC cell lines. In both H1836 and SHP-77 cells, *ASCL1* KD reduced the expression of both *PROX1* and *FOXA2* (Fig. 5h). Chromatin immunoprecipitation with sequencing (ChIP-seq) analysis confirmed direct binding of ASCL1 to the promoter region of *PROX1*, and to enhancer and promoter regions of *FOXA2*, in both SCLC cell lines (Fig. 5i, j). *PROX1* KD did not evidently affect the expression of *FOXA2* in either cell line (Fig. 5k), suggesting that *PROX1* scored as a differentially expressed gene in our analysis because, like *FOXA2*, it is a transcriptional target of ASCL1. Indeed, *SOX2* and *CACNA1A* which also came up in single-cell profiling (Fig. 5e) are proposed to be downstream targets of ASCL1[38]. *FOXA2* KD did not demonstrate a consistent effect on expression of either *ASCL1* or *PROX1* (Supplementary Fig. 5d). Taken together, these data suggest that ASCL1 regulates *FOXA2* expression in SCLC, and that ASCL1 might do so by directly engaging the transcriptional regulatory regions of *FOXA2*.

### ASCL1 binds the FOXA2 locus in met-associated SCLC

ASCL1 is expressed in most SCLC tumors regardless of metastasis and has been implicated as a necessary factor in SCLC oncogenesis[38]. Consistent with ASCL1 and FOXA2 having distinct roles, immuno-fluorescence (IF) analysis showed that ASCL1 and PROX1 were expressed in both never-met SCLC and in met-associated primary SCLC, whereas FOXA2 was enriched only in met-associated SCLC (Fig. 6a). Our initial bulk RNA-seq data also showed no significant differences in *ASCL1* or its suggested downstream genes (*PROX1*, *SOX2* and *CACNA1A*) expression despite the differences noted in *FOXA2* expression (Supplementary Fig. 6a). Although two ASCL1+ cell lines analyzed for ChIP-seq showed binding to FOXA2 locus (Fig. 5i), these observations suggested that while ASCL1 might promote *FOXA2* expression, an additional level of regulation must influence ASCL1

control of *FOXA2*. Indeed, scRNA-seq also demonstrated an ASCL1$^+$/FOXA2$^-$ cell population (Fig. 5g).

We further explored this mechanism in never-met primary SCLC vs. met-associated primary SCLC, by optimizing a protocol for ATAC-seq (assay for transposase-accessible chromatin using sequencing) applicable to formalin-fixed paraffin-embedded (FFPE) samples (Fig. 6b)[41,42]. The tumors processed for FFPE-ATAC analysis showed predominantly 3D intact nuclei as defined by DAPI staining (Supplementary Fig. 6b). This method showed accessibility to the internal control genes (*TUBA1A* and *ACTB*) supporting the quality of this assay (Supplementary Fig. 6c). These data suggested reduction in chromatin accessibility in the promoter and enhancer regions of the *FOXA2* locus in the never-met tumor relative to the metastatic-associated tumor after normalization (Peaknorm log$_2$FC = 2.47, P$_{adj}$ = 0.075) (Fig. 6c). Furthermore, comparison of the motif analysis in met-associated versus never-met primary samples showed enrichment of *FOXA2* binding sites in met-associated SCLC cases (Supplementary Fig. 6d). Limited *FOXA2* locus chromatin accessibility was evident in the never-met tumor, and we considered the possibility that this might arise from intratumoral heterogeneity of cell states. Consistent with this possibility, we did detect small areas of detectable FOXA2 expression in this sample by IF (Supplementary Fig. 6e).

To obtain a more definitive result, we also performed ATAC-seq analysis on an ASCL1$^+$ PDX tumor that was homogeneously negative for FOXA2, in comparison to 3 ASCL1$^+$/FOXA2$^+$ PDX tumors (Fig. 6d, Supplementary Fig. 6f). Supporting the primary tumor analysis, we found that the *FOXA2* locus was open in all 3 ASCL1$^+$/FOXA2$^+$ PDX tumors, but closed in the ASCL1$^+$/FOXA2$^-$ PDX tumor (Fig. 6e, Supplementary Fig. 6g). We generated primer sets spanning the promoter, enhancer, and a negative control region based on the *ASCL1* ChIP-seq data (Fig. 5i **and** Supplementary Data 7). Consistently, ChIP-qPCR showed significant enrichment of ASCL1 at the *FOXA2* regulatory regions in the three ASCL1$^+$;FOXA2$^+$ PDXs, but not in the ASCL1$^+$;FOXA2$^-$ PDX (Fig. 6f).

Given that not all ASCL1-positive SCLC express FOXA2, additional positive and/or negative regulators are likely to be involved in controling FOXA2 expression. Identification and characterization of these additional co-factors through detailed interrogation of ASCL1 co-factors or post-translational modifications, and of *FOXA2* gene regulatory elements and chromatin status, will be the focus of future analysis. Overall, these data suggest that differential chromatin accessibility at the *FOXA2* locus is one mechanism influencing ASCL1-dependent FOXA2 expression and metastatic competence in SCLC.

## Discussion

In this study, we identified FOXA2 as a determinant of metastatic competence in SCLC. As SCLC is known for its high metastatic potential, we hypothesized that exceptional SCLC patients whose disease presented with no evidence of spread even to local nodes, and who never suffered recurrence or metastasis throughout their clinical course after definitive local therapy might have a different biology than those with more typical metastatic SCLC. Specifically, such rare cases might be enriched for tumors lacking critical drivers of metastasis commonly expressed in more typical SCLC. Consistent with this hypothesis, never-met and met-associated primary tumors showed clear differences in gene expression profiles, defining a small set of transcriptional regulators specifically absent in never-met cases as putative drivers of metastasis. Among these, FOXA2 emerged as a leading candidate, and was further validated across multiple experimental models.

Despite a clear inhibitory effect of FOXA2 suppression on the metastatic activity of SCLC cells in mice, this had no apparent effect on SCLC viability, proliferation in vitro, or subcutaneous tumor growth in vivo. The specificity of this result stands in contrast to previously

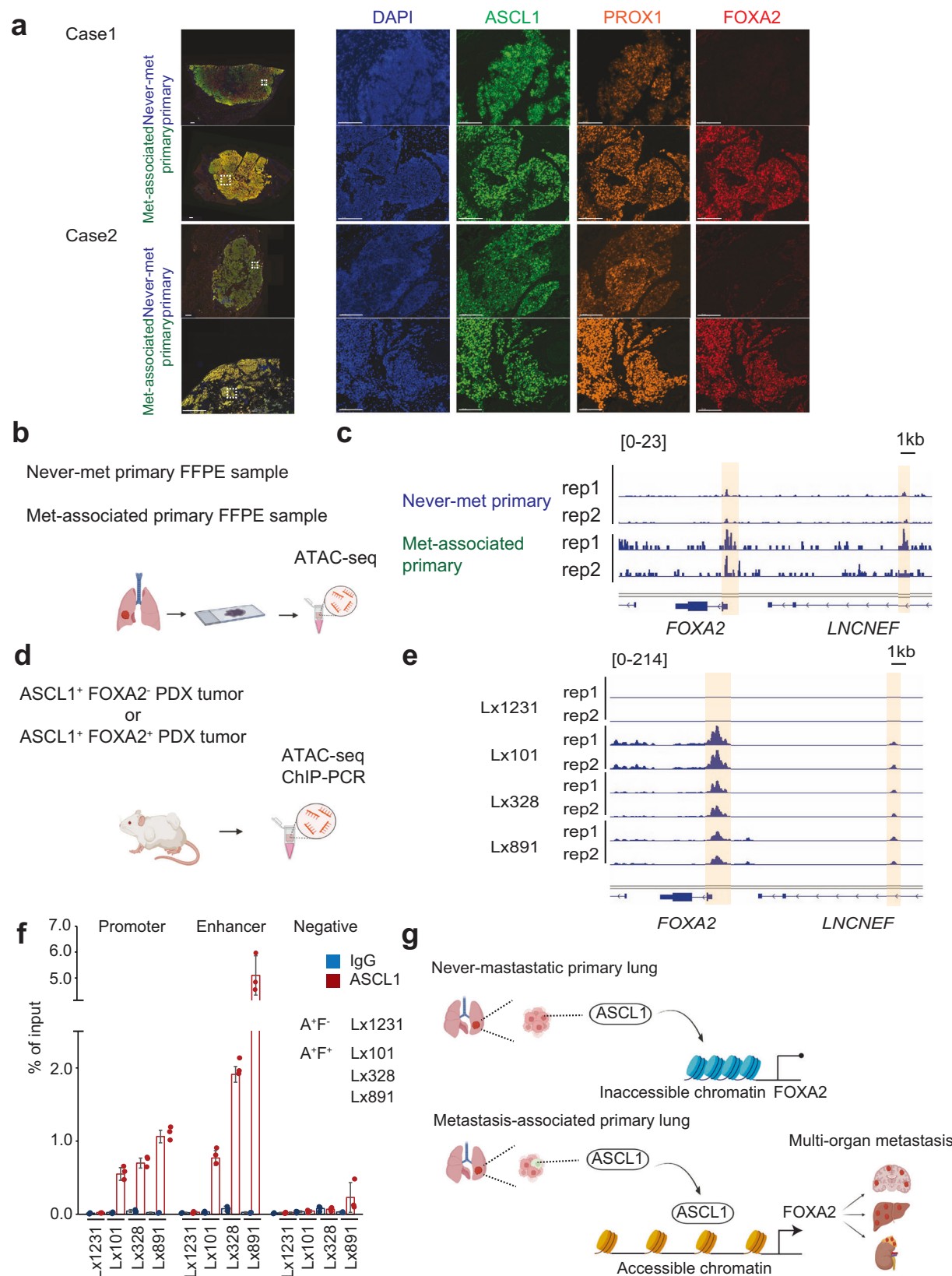

suggested metastatic drivers of SCLC that in fact have a dominant role in controlling SCLC growth[7,10,43]. *FOXA2* expression consistently induce a characteristic signature of fetal neuroendocrine gene expression, in line with the known role of *FOXA2* as a developmental regulator. The association of a fetal progenitor state as a precondition for metastasis is consistent with prior observations in LUAD, colon adenocarcinoma

and other solid tumors, suggesting a common metastatic competency requirement across cancer types[16,17]. Fetal gene signatures driving developmental pathways in tissue differentiation have been extensively implicated as key to establishment and expansion of metastases, across multiple tumor types (reviewed in ref. 20). Together these studies support that aberrant reactivation of early precursor pathways

**Fig. 6 | ASCL1 binds to the *FOXA2* locus in met-associated primary SCLC.**
**a** Immunofluorescence (IF) image of never-met primary and met-associated primary tumor clinical samples stained for DAPI, ASCL1, PROX1, and FOXA2 for 2 cases. Insets are indicated in white box. Scale bar: 1000 μm (overlay), 100 μm (DAPI, ASCL1, FOXA2). **b** Schematic of FFPE-ATAC-seq from never-met primary and met-associated primary tumor clinical samples. Created in BioRender. Rudin C. (2025) https://BioRender.com/mlreujw. **c** Chromatin accessibility assessed by ATAC-seq at the *FOXA2* locus (Peaknorm). Highlighted regions indicate promoter and enhancer domains. **d** Schematic of ATAC-seq and ChIP-qPCR on PDX tumors.

Created in BioRender. Rudin C. (2025) https://BioRender.com/mlreujw. **e** ATAC-seq comparing ASCL1⁺FOXA2⁺ PDX (Lx101, Lx328, Lx891) vs ASCL1⁺FOXA2⁻ (Lx1231) PDX tumors at the *FOXA2* locus. **f** IgG and ASCL1 ChIP-qPCR of the *FOXA2* locus, and negative control region in one ASCL1⁺FOXA2⁻ (A + F-) and three ASCL1⁺FOXA2⁺ (A + F + ) SCLC PDX models from panel. Bars are shown in Mean±S.D of $N = 3$ experimental replicates. **g** Schematic of ASCL1 interaction with *FOXA2* in never-met versus met-associated SCLC. Created in BioRender. Rudin C. (2025) https://BioRender.com/j34m841.

supporting normal tissue development enable metastasis precursors to survive, adapt, and proliferate in diverse microenvironments.

SCLC does differ from other solid tumors, such as breast cancer and colorectal cancer, in the characteristic pattern of early multifocal metastases. Our observational clinical data supports the aggressive multi-organ metastatic capacity of SCLC. In contrast to prior work using breast adenocarcinoma lines[25,26], reintroduction of metastatic foci of SCLC by intracardiac injection did not yield subclones enriched for differential organ-specific tropism. These data are consistent with clinical observations suggesting that the intrinsic metastatic competency of SCLC is typically sufficient to overcome multiple organ-specific barriers, and to provide an exceptional microenvironmental adaptability which facilitates diffuse metastatic spread. The multiple traits that have been described as required for metastatic competency[20] appear to be overcome in SCLC without a need for extensive organotropic adaptation upon dissemination.

We explored the upstream determinants of *FOXA2* expression in SCLC, identifying ASCL1 as a factor that upregulates FOXA2, and that directly binds to *FOXA2* gene promoter and enhancer elements. ASCL1 is a transcription factor essential for normal neuroendocrine differentiation in development. The murine homolog Ascl1 is also required for initial development of SCLC in the mouse lung[36]. That ASCL1 is implicated in the earliest steps of SCLC oncogenesis, and directly regulates expression of FOXA2, a potential driver of metastasis, offers an explanation for the early acquisition of metastatic competency in SCLC. However, ASCL1 expression in SCLC is more prevalent than FOXA2: both bulk and single cell analyses confirm the presence of ASCL1-expressing SCLC either with or without co-expression of FOXA2. In addition, while ASCL1 expression is a feature of at least 70% of SCLC, there are well-described subtypes of SCLC that lack detectable ASCL1 expression[37]. These observations have at least two important implications: first, FOXA2 is not the only factor that can promote metastatic competency in SCLC; and second, additional regulatory mechanisms beyond ASCL1 must control whether or not FOXA2 is expressed. It should be additionally emphasized that the ASCL1-FOXA2 axis is of relevance for NE-high subtypes of SCLC that express ASCL1. Other determinants of metastasis are likely to be of primary relevance in NE-low subtypes of SCLC.

To begin to address possible additional regulatory mechanisms, we analyzed chromatin accessibility as a correlate of FOXA2 expression in the context of ASCL1-expressing SCLC. ATAC-seq of ASCL1-expressing primary tumors and cell lines suggest differential chromatin accessibility of the *FOXA2* locus as one determinant of whether ASCL1 can induce FOXA2 expression (Fig. 6g). What factors might determine differential chromatin accessibility, and what additional positive or negative transcriptional regulators might affect FOXA2 transcription in accessible chromatin remain important questions for future investigation. Finally, the regulation of *FOXA2* expression by ASCL1 is a cancer cell-intrinsic mechanism; we have not assessed whether extrinsic signals from the tumor microenvironment could also impact this pathway or others, as has been shown to be the case for L1CAM, a key determinant of metastatic competency in lung, colorectal, and breast adenocarcinomas[17,44].

Recent observations in a genetically engineered mouse model of SCLC nominated FOXA1 and/or FOXA2 as candidate factors associated with SCLC metastasis[10]. The identification and characterization of FOXA2 as a contributor to metastatic competency in primary human SCLC is consistent with the predictions of the mouse model. We also note some key differences in apparent drivers of metastasis in human and murine SCLC—perhaps most notably NFIB, which is strongly associated with SCLC metastasis in the mouse; we have not observed differentially elevated *NFIB* expression in tumors with metastatic potential in our clinical datasets. Our data support FOXA2 as a candidate predictor of SCLC relapse in patients with definitively treated early stage disease.

In summary, we have identified the ASCL1-FOXA2 axis as a determinant of metastatic capacity in SCLC, and a potential contributor to the exceptionally early metastatic spread characteristic of this tumor type. Disruption of this axis, for example by target-specific degraders, offers an opportunity to reduce metastatic spread in SCLC, which could ultimately improve clinical outcome for patients with this highly lethal disease.

## Methods

All studies conducted as part of this work were reviewed by the institutional committee of Memorial Sloan Kettering Cancer Center (MSKCC). Clinical data and samples were analyzed under an Institutional Review Board approved protocol (14-091). Procedures for animals were performed under an approved Institutional Animal Care and Use Committee protocol (13-07-007).

### Clinical data acquisition

The clinical cohort described here are patients with SCLC whose tumors were sequenced using the MSK-IMPACT clinical targeted sequencing assay ($N = 327$) following informed consent. MSK-IMPACT data were collated for analysis in cBioportal. Please refer to Supplementary data 1 for further patient's characteristics.

### Animal procedures (intracardiac or subcutaneous injection of cell lines and PDX), and animal measurements (BLI)

Female NOD.Cg-Prkdc<scid> Il2rg<tm1Wjl > /SzJ (NSG) mice which were aged matched between 6 and 10 weeks were used for each experiment. The number of mice used in each experiment are shown in figures or in figure legends. Metastasis assays were performed with intra-cardiac (i.c.) inoculation to the mice using $5.0 * 10^4$ cells (for cell lines) to $5.0 * 10^5$ cells (for PDX, FOXA2 KD, mFoxa2 OE experiments) resuspended in 0.1 ml PBS and injected into the left ventricle of the mice. Measurement of the luciferin signals were performed every week by IVIS. Calculation of the signals were performed using the software "Living Image 4.2 (Perkin Elmer)". Collections of the tumors were performed at 1 month (H82), 2 month (H1836, H1963), and 2–3 month (PDX) with reference to protocol-defined ethical criteria for euthanasia. PDX tumors (Lx773I, Lx599B, Lx101, Lx328, Lx891, Lx1231) were generated by injecting $2.0 * 10^6$ cells/ tumor to the mouse flank. Tumors were collected when they reach to 600–800 mm³ (tumor volume calculated as [width² ×length]/2) following the rule of maximum tumor size permitted by the institutional committee (1000 mm³). Subcutaneous tumor measurements were performed at least weekly. Doxycycline-containing chow(2 g/kg) was given ad libitum starting 1 week prior to i.c. injection. Mice were consistently

housed and controlled under the environmental conditions of: $21 \pm 1.5\,°C$ temperature, $55 \pm 10\%$ humidity and a 12 h light–dark cycle (lights were on from 6:00 to 18:00). Mice were euthanized using $CO_2$ in the following situations: tumor volume exceeding $1000\,mm^3$, weight loss >10%, poor grooming, hunched posture, distended abdomen, tumor ulceration, or lack of mobility following the recommendations of the American Veterinary Medical Association (AVMA).

### Cell lines

H1836 (ATCC, Cat #:CRL-5898), SHP-77 (ATCC, Cat #:CRL-2195), H82 (ATCC, Cat #:HTB-175), and H1963 (ATCC, Cat #:CRL-5982) were purchased from ATCC. Cell lines were authenticated by STR verification and regularly tested for mycoplasma (Universal Mycoplasma Detection Kit, ATCC). H1836 was maintained in DMEM/F12 HITES media 10% FBS and SHP-77, H82 and H1963 were maintained in RPMI-1640 10% FBS as recommended by ATCC. Genetic inhibition in shRNA experiments was induced by treating with doxycycline $1\,\mu g/mL$.

### Gene manipulation (vector construction and virus production, transduction)

Dox-inducible shFOXA2 and shPROX1, and constitutive shASCL1 (Millipore Sigma Cat #:TRCN0000013551, TRCN00000244309) were used for the assays. ShRNA predictions were generated by Splash (http://splashrna.mskcc.org/) using NCBI Gene IDs. 97mer shRNA oligos were synthesized, PCR amplified, and cloned into a mirE lentiviral dox-inducible backbone with GFP/RFP and Puro (MSK Gene Editing & Screening Core with permission from Johannes Zuber Lab). Ligations were transformed using Mach 1 Competent Cells (Thermo Cat:#C862003) and DNA was extracted from single colonies by miniprep (Qiagen). Positive clones were identified by Sanger sequencing. ORF Clone to human FOXA2 (NM_021784.4) (Genscript;ORF Clone OHu31655) was PCR amplified to incorporate restriction sites for cloning. Lenti-Cas9-2A-EGFP plasmid was amplified and Cas9 was removed by digestion and religation. PCR-amplified ORF was digested and restriction cloned into Lenti-2A-EGFP. For the generation of mouse Foxa2 overexpression plasmid vector, pCW-2A-EGFP was cut with MluI/XbaI and 6.5 Kb fragment was gel extracted. Mouse Foxa2 ORF (CCDS 71157.1; 1398 nt) and T2A-TurboRFP-hPGK-BSD fragment were synthesized as gene fragments (Twist Biosciences) and Gibson cloned with extracted 6.5 Kb fragment to generate pCW-mFoxa2-T2A-TurboRFP-hPGK-BSD. Cloning was sanger verified. Ligations were transformed using Stbl3 Competent Cells (Thermo Cat:#C737303) and DNA was extracted from single colonies by miniprep (Qiagen). Positive clones were identified by Sanger sequencing. Lentiviral particles were produced by transfecting HEK293T cells (ATCC, Cat #: CRL-1573) with the vector of interest together with pMD2.G (Addgene, Cat #:12259) and psPAX2 (Addgene, Cat #:12260) packaging vectors (3:2:1 ratio of plasmid of interest: psPAX2:pMD2.G) and with JetPrime transfection reagent (Polyplus)[32]. Lentivirus was collected 72 h after transfection and concentrated 1:20 with Lenti-X™ Concentrator following manufacture's protocol (Takara Bio Cat#:631232). The lentivirus particles were transduced into cell lines using polybrene (8 mg/ml), and selected after 5–7 days with a corresponding antibiotic. The specific sequence details are listed in the Supplementary Data.

### Colony formation and proliferation assays

Colony formation assays were performed using 100,000 cells (H1836), 75,000 cells (SHP77), and 400,000 cells (H1963). These cell lines were pre-treated for 5 days with Doxycycline $(1\,\mu g/mL)$ to induce sufficient FOXA2 KD. Each well (>3 wells for each condition) were pre-coated with 2.8% agarose. Cell seeding was performed with single-cell dissociated cells in 1.4% agarose with media. After these agarose layers are solidified, normal media (RPMI-1640 with 10% FBS media or DMEM/F12 HITES with 10% FBS media) with Doxycycline was added on the top of the wells. Doxycycline was added from day1 and replenished every 48 hr. The images were taken using an Oxford Optronix Gelcount instrument and the number of formed clusters was counted.

Cell proliferation assays were performed with 1000 cells/well (SHP-77) and 2000 cells/well (H1836) seeded in 96 well-plates. The cells were treated with doxycycline starting on day 4 at $1\,\mu g/mL$ and replenished every 48 hr through day 10. Luminescence was measured using the CellTiter-Glo 2.0 Assay (Promega, G9242) following the manufacturer's instructions. The proliferation rate was calculated by normalizing to day 1.

### RNA isolation from FFPE samples

The FFPE samples were first evaluated by a thoracic pathologist to confirm tumor involvement. Subsequently, 10−20 unstained sections (each 10 thickness) were prepared for RNA extraction for each sample. The sections were deparaffinized using $800\,\mu L$ of mineral oil (Fisher Scientific, Cat #: AC415080010) and $180\,\mu L$ of Buffer PKD. This procedure was followed by the addition of proteinase K to facilitate tissue digestion. The mixture was incubated at 56 °C for 15 min. Centrifugation was used to promote phase separation, after which the aqueous phase was chilled for 3 min to precipitate RNA. Following a 15-min centrifugation at $20,000 \times g$, the RNA-containing supernatant was collected for extraction. Nucleic acids were purified using the AllPrep DNA/RNA Mini Kit (QIAGEN, Cat #: 80204) according to the manufacturer's protocol. Finally, RNA was eluted in nuclease-free water, and DNA was eluted in 0.5× Buffer ATE.

### RNA sequencing and pathway enrichment analysis by GSEA

Approximately 500 ng of FFPE RNA was used for RNA library construction using the KAPA RNA Hyper library prep kit (Roche) following the manufacturer's instructions with minor modifications. Sample-specific dual-index primers (Integrated DNA Technologies) together with customized adapters with unique molecular indexes (UMI; Integrated DNA Technologies) were added to each library. The quality of the libraries was assessed by the TapStation Genomic DNA Assay (Agilent Technologies), and its quantity of libraries was measured with Qubit (ThermoFisher Scientific). Equal amounts of each RNA library were pooled for hybridization capture with IDT Whole-Exome Panel V2 (Integrated DNA Technologies) using a customized capture protocol modified from the NimbleGen SeqCap Target Enrichment system (Roche). Sequence was performed on an Illumina HiSeq4000 with paired end reads $(2\,Å \sim 100\,bp)$, at 50 million reads/sample. Sequencing for clinical samples was performed in Integrated Genomics Operation Core (MSKCC) and the cell line samples were sequenced in Genewiz. Fastq files were mapped to the human genome (hg38) and read counts per gene were quantified using STAR[45] with default parameters and genecode (v28) annotation file. Differentially expressed genes (DEGs) were identified with DESeq2[46], with a fold change cutoff of 1.5 and FDR < = 0.01.

Pathway enrichment analysis was performed using Gene set enrichment analysis (GSEA, v4.0.2)[47] using ClusterProfiler R package v3.18.1[48]. To extract the candidate pathways, differential expression analyses between FOXA2 KD (or FOXA2 OE) and control(wild) cells was performed (met-associated primary lung vs never-met primary for clinical samples) on the full set of genes ranked by p-value scores computed as $-\log10(p\text{-value})*(\text{sign of } \log2FC)$. Gene set annotations were obtained from Molecular Signatures Database (MSigDB v7.0.1)[47,49] and enrichment was calculated by using permutation test with p-value adjustment by Benjamin–Hochberg procedure. Enriched gene sets with adjusted p-value ≤ 0.05 were regarded as significant. Pathways and gene sets of interest were identified using the GSEA-msigdb database (https://www.gsea-msigdb.org/gsea/index.jsp) and Gene Ontology resources (http://geneontology.org/). Pathways were highlighted based on the respective normalized enrichment scores (NES) for each cell line.

## Immunohistochemistry (IHC)

IHC was performed using the Leica/BOND system. HIER pretreatment was done with ER2 retrieval solution and deparaffination was done using BOND Demex solution (AR9222). The primary antibodies used were ASCL1 (Invitrogen, Cat#:24B72D11, 1:100), NEUROD1 (Abcam, Cat#:ab205300, 1:100), POU2F3 (Santa Cruz, Cat #: sc-293402, 1:1500), Ki67 (Abcam, Cat #: ab16667, 1:1000), and FOXA2 (Abcam, Cat #: ab108422, 1:1000). The detection was done using the BOND Polymer Refine Detection System (DS9800).

For TMA H-score, the expression was scored in a blinded manner by pathologists, whereby the optical density level ("0" for no brown color, "1" for faint and fine brown chromogen deposition, "2" for intermediate chromogen deposition and "3" for prominent chromogen deposition) was multiplied by the percentage of cells at each staining level, resulting in a total H-score range of 0–300.

## Immunofluorescence (IF)

Automated multiplex IF was performed using Leica Bond BX staining system. First, FFPE tissue section was prepared in 5 µm. The sections were baked at 58 °C for 1 h prior to staining, loaded in Leica Bond, and dewaxed and pretreated with EDTA-based epitope retrieval ER2 solution (Leica, Cat#: AR9640) for 20 min at 95 °C. A sequential 3-plex antibody staining and detection protocol was used using the primary antibodies to FOXA2 (10ug/ml, 1:200, Rb, Abcam, Cat#:ab108422), Mash1 (0.3125ug/ml,1;1700, Rb, Abcam, Cat#:ab211327), and PROX1 (0.3ug/ml, 1:500, Rb, Cell Signaling Technology, Cat #:14963S). These slides were incubated for 1 hr at RT followed by incubation with Leica Bond Polymer anti-rabbit HRP secondary antibody for 8 min. This staining was performed automatically using the manufacturer's instructions of Leica Kit (Cat#: DS9800). Signal detection was achieved through the use of CF® dye tyramide conjugates (Biotium, Cat#: 92174) or Alexa Fluor tyramide signal amplification reagents (Life Technologies, Cat#: B40953, B40958). After each round of IF staining, Epitope retrieval was performed for denaturization of primary and secondary antibodies before another primary antibody was applied. After the run, slides were washed in PBS and incubated in 5 µg/ml 4',6-diamidino-2-phenylindole (DAPI) (Sigma Aldrich) in PBS for 5 min, rinsed in PBS, and mounted in Mowiol 4–88 (Calbiochem). Slides were kept overnight at −20 °C prior imaging and it was scanned on a Panoramic Scanner (3DHistech) using a 20x/0.8NA objective.

## Western blot

Cell pellets were lysed with cold RIPA buffer (Thermo Scientific), sonicated, and incubated on ice for 20' followed by centrifugation at 10,000 rcf at 4 °C for 10 min. Protein quantification was performed using a Pierce™ BCA Protein Assay kit (Thermo Scientific). Antibodies for Western blotting included FOXA2 (Abcam, Cat #: ab108422), ASCL1 (Cell Signaling Technology, Cat #:10585S), PROX1 (Cell Signaling Technology, Cat #:14963S), and GAPDH (Cell Signaling Technology, Cat #: 97166S).

## ChIP-seq and ChIP-qPCR

Two cell lines (H1836, SHP-77) and 4 PDX tumors (Lx1231, Lx891, Lx328, Lx101) were prepared for ChIP-seq analysis in number of $1 \times 10^7$ cells. First, the samples were fixed at room temperature with 1% formaldehyde (Sigma Aldrich) and quenched by addition of 125 mM glycine. Cells were resuspended with ChIP lysis buffer supplemented with phosphatase inhibitors (Thermo Scientific, Halt Phosphatase Inhibitor Cocktail, Cat #: 78427, 1:1000) and protease inhibitors (Roche cOmplete mini EDTA-free protease inhibitor tablets, Cat #:11836170001) after PBS wash. Cell lysates in ChIP lysis buffer were sonicated and its supernatant were incubated with the antibodies at 4 °C overnight (IgG: Cell Signaling Technology, Cat #: 2729S (1 mg/ml) and ASCL1: Cell Signaling Technology, Cat #: 43666S, (100 µg/ml)). Each antibody is used in 0.2 µg/ $1 \times 10^6$ cells). At the same time, Dynabeads™ Protein G were blocked with 2.5 mg/mL UltraPure™ BSA (Thermo Scientific, Cat #: AM2616) at 4 °C overnight. Chromatin-antibody samples were incubated with the beads which were washed with ChIP buffer for 3 times. After wash with high salt ChIP buffer and Tris-EDTA (TE) buffer, chromatin complexes were eluted in TE with 1% ultrapure SDS (ThermoFisher Scientific). Finally, the samples were reverse-crosslinked with RNAse A (ThermoFisher Scientific, Cat #: EN0531), followed by incubation with Proteinase K solution (ThermoFisher Scientific, Cat #: EO0491). DNA was purified using QIAquick PCR Purification Kit (Qiagen) and sequenced by Illumina sequencing libraries which was prepared with KAPA HTP Library Preparation Kit (Kapa Biosystems, Cat #: KK8234) according to the manufacturer's instructions with 0.2-5 ng input DNA and 8-14 cycles of PCR. An average of 20-30 million paired reads were generated per sample.

For the ChIP-qPCR analysis using PDX tumors, the samples were processed in the same manner as above. PDX tumors collected from the mouse flank were dissociated into single cells, and fixated. DNA samples immunoprecipitated following the ChIP protocol were analyzed by qRT-PCR, and the amplification product was expressed as percentage of the input. Information of qPCR primers are listed in Supplementary Data.7.

## Sample preparation for scRNA-seq

Clinical sample preparation for single-cell RNA-sequencing were mechanically/enzymatically dissociated using the tumor dissociation kit (Miltenyi, Cat #:130-095-929,) and the Gentle MACS Octo Dissociator with Heaters (Miltenyi, Cat#:130-096-427)[32,50]. The samples were processed for dissociation and filtered with MACS Smart Strainers (70 mm) (Miltenyi, Cat#: 130-098-462). The materials were FACS-sorted with DAPI (Thermo Fischer Scientific, Cat#:D1306), Calcein AM (Biolegend Cat#: 425201), PE anti-CD45 antibody (working dilution 3uL/100uL, Biolegend, Cat#: 368510) and processed for scRNA-seq analysis (gating strategy is shown in Supplementary Fig. 5b).

## ATAC-seq and data analysis (PDX and FFPE samples)

To generate the data for FFPE-ATAC-seq, FFPE material in 20 µm was submitted at the MSKCC's Epigenetics Research Innovation Lab for processing. FFPE cell isolation was performed with FFPE Tissue dissociation kit (Miltenyi #133-118-052) following manufacturer's instructions. Upon isolation, FFPE -derived cells were prepared for imaging in order to estimate 3D intact architecture (doi.org/10.1371/journal.pone.0223759). To perform the imaging, glass coverslips were coated with poly-L-lysine. Coverslips were incubated in a 0.1% poly-L-lysine solution for a minimum of 15 min, air-dried and placed into a 24-well plate. FFPE-derived cells were dropped onto the polylysine coated coverslips ( ~ $3 \times 10^5$ cells in a 30 µl drop of 1X PBS) and left to sediment for about 10 min. The collected cells were centrifuged and washed with 1 x PBS, permeabilized in 0.5% Triton-X/1x PBS and rinsed twice with 1 x PBS. Coverslips containing cells were then stained with DAPI and mounted. Images were acquired using Nikon Eclipse Ti V5.20microscope unit with an Andor Zyla VSC-01979 camera, and optical sections were captured with a 60x objective. Cells were then analyzed and examined with Fiji (doi: 10.1038/nmeth.2019) for intact 3D architecture (as defined by DAPI staining and particularly focusing on the nuclei edges). For both FFPE samples under investigation, more than 100 nuclei were analyzed in order to evaluate 3D intact morphology. Viability was estimated to be >90% (Supplementary Fig. 6b).

For PDX ATAC-seq, tumors were collected from the mice subcutaneous area when they reached to ~600–800 mm³. These tumors were processed and collected as single cells. Approximately, 60,000 cells were prepared for FFPE-ATAC-seq and PDX-ATAC-seq. ATAC-seq was performed using the Tagment DNA TDE1 Enzyme (Illumina, Cat #:20034198) following the manufacturer's instructions[41,42]. Nuclei were

isolated with centrifugation (10 min at 800 g at 4 °C) followed by the addition of 50 μL transposition reaction mix (25 μL TD buffer, 2.5 μL Tn5 transposase and 22.5 μL ddH$_2$O) using Tagment DNA TDE1 enzyme. Samples were then incubated at 37 °C for 35 min. DNA was isolated using a ZYMO Kit (D4014). ATAC-seq libraries were prepared using NEBNext High-Fidelity 2X PCR Master Mix (NEB, Cat #:M0541) using a uniquely barcoded primer per sample, and a universal primer. Sequencing libraries were sent to the Integrated Genomics Operation Core (MSKCC) for sequencing on a NovaSeq 6000. Raw sequencing reads were trimmed and filtered for quality ($Q > 15$) and adapter content using version 0.4.5 of TrimGalore (https://www.bioinformatics.babraham.ac.uk/projects/trim_galore) and running version 1.15 of cutadapt and version 0.11.5 of FastQC. Version 2.3.4.1 of bowtie2 (http://bowtie-bio.sourceforge.net/bowtie2/index.shtml) was employed to align reads to mouse assembly mm10 and alignments were deduplicated using MarkDuplicates in Picard Tools v2.16.0. Enriched regions were discovered using MACS2 (https://github.com/taoliu/MACS) with a $p$-value setting of 0.001, filtered for blacklisted regions (http://mitra.stanford.edu/kundaje/akundaje/release/blacklists/mm10-mouse/mm10.blacklist.bed.gz), and a peak atlas was created using +/− 250 bp around peak summits. The Normalized bigwig files were created using BEDTools suite (http://bedtools.readthedocs.io). For the analysis, peak-gene associations were created by assigning all intragenic peaks to that gene, while intergenic peaks were assigned using linear genomic distance to transcription start sites (TSS). Calculation of differential enrichment for all pairwise contrasts was performed on raw counts matrix and DESeq2 build by featureCounts Version 1.6.1 (http://subread.sourceforge.net). Pathway enrichment was calculated by assigning each gene a unique score based on the associated peak with the greatest magnitude change and running GSEA in pre-ranked mode. Motif signatures were obtained using Homer v4.5 (http://homer.ucsd.edu) on differentially enriched peak regions.

## Single cell RNA-seq methods

**Pre-processing.** All analyses were done in Python using Scanpy[51] (v1.10) unless otherwise specified. Raw FASTQ sequencing files were aligned to the reference genome version GRCh38, barcodes were filtered, and unique molecular identifiers (UMI) were identified using 10x Cell Ranger software (v6.0.1). Cell by gene matrices were loaded as anndata objects. Cells with <200 genes or >40% mitochondrial content were removed. Doublet cells were removed using scrublet[52]. Highly variable genes were selected using the highly_variable_genes function with the flavor option set to seurat_v3. The raw counts were then normalized by library size and log-transformed with a pseudo-count of 1. Principal component analysis (PCA) was performed retaining the top 50 PCs. Harmonypy[53] (v0.0.9) was used to perform batch correction. The batch corrected embeddings were used for leiden clustering and creating the UMAP [McInnes] projection for visualization.

**Major cell compartment annotation.** The major cell components were defined using reference-based mapping[54] in Seurat (v5.0)[55]. Briefly, cells were exported to Seurat objects and mapped to the human lung cell atlas[56] using the RunAzimuth() function. This yielded multi-level annotations. Level 1 annotation was used to denote the major compartment for each cell.

The above procedure was checked against the following procedure, which gave highly concordant results: the major cell compartments were defined using the markers for each component. Each cell was scored for each of the main compartments using the function score_genes. Then each cell was assigned to the cell compartment with the highest score.

**Identifying malignant cells.** To identify malignant cells in the epithelial compartment, single-cell copy number signal was computed per sample using InferCNV (v1.14)[57],utilizing a reference set of cells from all other samples (five cells per sample across major cell compartments: epithelial, stromal (fibroblast), immune and endothelial). Copy number signal was then re-binned to 10 Mb sized bins, prior to performing PCA to 50 dimensions and louvain clustering over a nearest neighbor graph with cosine distance. Clusters were manually inspected for each patient to assign malignant cell clusters, which were based near total enrichment of epithelial cells harboring distinct copy number profiles in clusters.

**Identifying genes associated with FOXA2 expression.** To identify genes associated with the expression of FOXA2, a cohort of 19 patients with sufficiently high expression of FOXA2 were assembled. Sufficiently high expression was defined as comprising at least 10% FOXA2 expressing cells in >200 malignant cells. Generalized linear models were used to model the expression of FOXA2 given 7000 highly variable genes. Gene expression counts were normalized for library-size, log-transformed, and finally standardized such that the mean and the variance of each gene are zero and one respectively. Using regression with an L2 penalty a per patient coefficient vector of per gene association with FOXA2 expression was estimated. The collection of these coefficients was used to compute a Z-score per gene and use it to rank the genes associated with FOXA2 expression in malignant cells.

**Pathway analysis.** Malignant cells were divided into two groups: FOXA2+ and FOXA2- with those with the raw count greater than zero in the former and the rest in the latter. Markers for cell types in the human fetal lung were acquired from He P et al.[35]. For each gene set, all malignant cells were using the function score_genes. Mann–Whitney $U$ test was used to compute a $p$-value.

**Epigenomic analysis.** Raw reads were processed using the same pipeline described in the ATAC-seq section. Enriched binding regions were called against the input or IgG reference samples using MACS2 with $p$-value < 0.001[58]. The genomic 'blacklisted' regions (http://mitra.stanford.edu/kundaje/akundaje/release/blacklists/hg38-uman/hg38.blacklist.bed.gz) were subsequently filtered. The filtered peaks within 500 bp were merged to create a union of peak atlas for either input or IgG as references. Raw read counts were tabulated over this peak atlas using featureCounts v1.6.0[59]. The read counts were then normalized with DESeq2. The read density profile in the format of bigwig file for each sample was created using the BEDTools suite (https://bedtools.readthedocs.io) with the normalization factor from DESeq2[46]. ChIP-seq data was visualized with the Integrative Genomics Viewer (IGV).

## Statistical analysis

To obtain an estimate of the optimal H score cutpoint for recurrence (23.5), we used the 'cutpointr' R package with the parameters method = maximize_metric and metric = sum_sens_spec using FOXA2 H-score from the TMAs as the predictor and recurrence as a binary outcome[60].

Comparisons between two groups were performed using two-tailed Student's $t$-test, as indicated in Figure legends. For three or more groups, one-way ANOVA and Wald's test were used. For TMB calculation, we used Mann–Whitney U test. All bars within the graphs represent mean values, and the error bars represent SEMs or standard deviation, as indicated in the Figure legends. $P$-values <0.05 were considered as significant.

## Statistics and reproducibility

All in vitro experiments were repeated a minimum of twice independently with similar results (western blots for 3 times (Figs. 3k, 5h and 5k)). For FOXA2 KD in cell experiments, two cell lines and two shRNAs targeting distinct sequences of the gene of interest were used. For mouse tumor analysis, intra-cardiac injection were conducted in 2 shRNA, 2 cell lines and >2 times showing consistent results. For epigenetic analysis, IgG control and ASCL1 ChiP-seq, and ATAC-seq were performed in two

replicates. ChiP-qPCR were performed in 3 replicates. Micrographs are confirmed for the following times as an individual experiment; Fig. 3i(twice), 5a and 6a (once in IHC and once in IF in 3 independent patient samples). No statistical method was used to predetermine the sample size.

## Reporting summary

Further information on research design is available in the Nature Portfolio Reporting Summary linked to this article.

## Data availability

Raw sequencing reads and processed files for RNA-seq, ChiP-seq, ATAC-seq and scRNA-seq are deposited in Gene Expression Omnibus database (GEO) under the accession number GSE281523,GSE281524,GSE281525, GSE281740. Source data are provided with this paper.

## Code availability

CODE is uploaded to github [https://github.com/shahcompbio/SCLC_MET.git] [https://doi.org/10.5281/zenodo.15257973][61].

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

## Acknowledgements
We acknowledge support from the Cancer Data Science Initiative (CDSI), the MSK Pathology Core, Gene Editing & Screening Core, Molecular Cytology Core, Animal Imaging Core, Single Cell Analytics Innovation Lab, and the Integrated Genomics Operation Core. We also acknowledge support from John Philip and Nadia Bahadur for obtaining clinical data with the MSK Extract CCED platform and technical support from Ioannis Karagiannidis (Weill Cornell Medicine, New York) for imaging 3D-intact nuclei. Figure panels (1b, 2a, 3j, 6b, d, g, Supplementary Fig. 1c) are generated with Biorender. This study was supported by National Cancer Institute grants R35-CA252978 (J.M.), R35-CA263816, U24-CA213274, and P30-CA008748 (MSKCC); the Druckenmiller Center for Lung Cancer Research (K.K., E.R., A.Q.V. and C.M.R.); the Alan and Sandra Gerry Metastasis and Tumor Ecosystems Center (J.M. and J.L.); the Robert J. and Helen C. Kleberg Foundation (C.M.R.); and the Van Andel Research Institute—Stand Up to Cancer Epigenetics Dream Team (C.M.R.).

## Author contributions

K.K., J.M. and C.M.R. designed the study and primarily interpreted the results. K.K. performed experiments and analyzed data; bioinformatic analysis was done by S.S., Y.A.Z., S.E.T., N.C., M.Z., E.H., P.J.H., R.P.K., A.M., S.P.S.; J.H.L., E.S. performed sample processing for epigenetic experiments, RNA extraction of FFPE samples were done by F.M. and B.L.; K. C., H.Z., E.R., E. de S. performed in vivo mouse experiments; Á.Q.V. provided strategic advice and assisted with project oversight; P.M., D.K., B.P.M., H.S. provided technical assistance; IHC and pathological analysis was performed by I.L. and U.K.B.; clinical cohort data were provided by N.J. S.; manuscript preparation, review and editing are done by K.K., J. M., C.M.R.; All authors have read and approved the final version of the manuscript.

## Competing interests

A.Q.V. has received honoraria from and is currently fully employed by Astra Zeneca. J.M. holds company stock from Scholar Rock. C.M.R. has consulted regarding oncology drug development with AbbVie, Amgen, Astra Zeneca, Boehringer Ingelheim, Daiichi Sankyo, Genentech/Roche, Jazz, and Merck, and serves on the scientific advisory boards of Auron, Bridge Medicines, DISCO, Earli, and Harpoon Therapeutics. C.M.R. has received licensing fees and royalties based on DLL3 antibody-based therapeutics. The remaining authors declare no competing interests.
