## [Transparent Peer Review file · Nature Communications]

FOXA2 promotes metastatic competence in small cell lung cancer

Corresponding Author: Dr Charles Rudin

Version 0:

Reviewer comments:

Reviewer #1

(Remarks to the Author)

Kawasaki K et al identifies ASCL1-FOXA2 axis as a key driver of metastatic competence in small cell lung cancer (SCLC). FOXA2 promotes a fetal neuroendocrine gene expression program that enables multi-organ metastatic capacity without impacting tumor proliferation. Tumors lacking FOXA2 ("never-met") show distinct gene expression profiles, with non-genetic mechanisms being critical to metastatic potential. In preclinical models, FOXA2 knockdown significantly reduced metastatic burden, while chromatin accessibility at the FOXA2 locus influenced its regulation by ASCL1. Clinically, FOXA2 expression is associated with poor relapse-free survival, highlighting its potential as a biomarker and therapeutic target for SCLC metastasis.

Strengths:

The study tackles a critical issue by investigating the drivers of metastasis in small cell lung cancer (SCLC), an area of significant clinical relevance. The authors take an elegant approach by comparing "never-metastatic/recurrent" SCLC cases with metastatic SCLC, providing a clear framework to study the biological drivers of metastasis. It is intriguing that SCLC metastases lack strong selective organotropism, which differentiates it from other cancer types.

Areas for improvement:

1. While the experimental design is original and results are robust, the novelty of the findings is not obvious as ASCL1-FOXA2 has previously been linked to drive SCLC metastasis (<https://doi.org/10.1158/0008-5472.CAN-23-1079>). Furthermore, the authors note that ASCL1 expression is far more prevalent than FOXA2, and SCLC exists with or without co-expression of FOXA2. This implies that ASCL1 likely drives a broader transcriptional program beyond FOXA2. Characterizing this larger ASCL1-driven program could be more impactful than focusing on FOXA2 alone. Additionally, ASCL1-high/FOXA2-low cases exist, suggesting that there are additional, unidentified regulators of FOXA2. These factors should be explored to complete the story.

2. ASCL1-FOXA2 Axis Validation:

a. Does ASCL1 knockdown abolish FOXA2 expression and metastatic propensity in metastatic SCLC models? The authors show in Fig 5h, that indeed knockdown of ASCL1 decreased FOXA2 protein levels in H1836 and SHP-77 cell lines, the effect on metastatic propensity seems to be not included. The authors should include these results.

b. Does the ASCL1-FOXA2 regulatory axis only apply to ASCL1-positive SCLC subtypes? Would FOXA2 KD in none ASCL-1 subtypes have no impact on metastasis?

c. Are all FOXA2-high tumors also ASCL1-high?

3. To further functionally validate the ASCL1-FOXA2 Axis:

a. The authors should explore if (low-moderate) overexpression of FOXA2 and/or ASCL1 in non-metastatic SCLC is sufficient to drive metastasis. This would provide stronger evidence for the role of the ASCL1-FOXA2 Axis as drivers of SCLC metastasis.

b. Specifically, does overexpression ASCL1-FOXA2 Axis lead to increased metastatic potential or penetrance, and does it preferentially promote liver metastasis? This is particularly relevant since shFOXA2 greatly suppresses liver metastases (Extended Data Fig. 3e), while its effects on adrenal and ovarian metastases are less pronounced. This hints at potential organotropism, mediated by FOXA2 which would be conceptually very interesting.

4. What is the organotrophic profile of metastases in FOXA2-high patients? Is there a liver preference in these cases, consistent with preclinical findings?

5. Were there any differences in TMB and could this influence metastatic potential between never-metastatic and metastatic SCLC cases. Might be a point to discuss or address as it should be available within the current datasets.

Minor Comments:

1. Many graphs are missing y-axis ticks, making interpretation difficult.

2. Figures 3 and 4 could be combined into a single figure for clarity and better coherence with the rest of the figures. Overall the figures would do well with a uniform standardization.

3. Figure legends would benefit from declarative titles that summarize key findings (e.g., "FOXA2 Knockdown Reduces Liver Metastases").

Reviewer #2

(Remarks to the Author)

This study makes the interesting finding from human tumor data from SCLC patients that the FOXA2 level, a transcription factor, is enriched in primary tumors of patients with metastatic disease compared to those who have not progressed with metastasis. This is demonstrated from a rich data set from MSKCC that includes clinical outcomes, whole genome sequencing, and RNAseq, and is shown clearly. The importance of FOXA2 for metastatic capability of the SCLC cells is demonstrated with experiments knocking down levels of FOXA2 in SCLC cell line models and testing them for the efficiency of forming metastasis in immune compromised mice. These data would be even more compelling with complementary experiments with overexpression in a FOXA2 low cell line and performing similar assays. The authors go on to identify ASCL1 as a possible regulator of FOXA2, a relationship that has been previously reported for SCLC. Importantly, they note this cannot be a simple one to one relationship since there are tumor samples expressing ASCL1 but low or no FOXA2. To understand what other mechanisms are at play, they use ATAC-seq of SCLC PDX that express or don't express FOXA2 and conclude chromatin accessibility is involved. The latter finding is a circular argument and is probably overinterpreted. See below for specific comments.

1) The knock down of FOXA2 showing loss of metastatic efficiency in 2 SCLC cell line models is the important experiment shown supporting one of the main conclusions in the study, that being the requirement for FOXA2 for metastasis in SCLC. From what is shown, the results are clean and dramatic. However, the data should be shown more thoroughly. For example, data points for the individual mice should be shown on the graphs in Fig. 3c-f. Some of the extended Fig. 3 data (such as extended Fig. 3c-e) could be moved into the main figure due to the importance of these experiments. Additional experiments shown in Fig. 4a-f showing FOXA2 is not required for xenograft tumor growth also need individual animals shown.

2) As stated above, the most compelling finding is the apparent requirement for FOXA2 in the in vivo metastasis test using SCLC cell lines introduced intracardially in mice. An important complement to these findings is testing a SCLC cell line with low FOXA2 normally but is engineered to exogenously express FOXA2-- tested for liver metastasis as per the assay used here. The cells have already been developed and used in RNA-seq experiments in his study. Another experiment that would significantly strengthen the arguments for the role of FOXA2 in metastasis is manipulating levels of FOXA2 in the PDX FOXA2 high and low models and testing occurrence of mets.

3) What happens to ASCL1 levels with FOXA2 KD? Is there a feedback relationship? FOXA2 as a pioneer factor may increase ASCL1 levels consistent with the IFs of tissue samples shown in Fig. 6.

4) The ATAC data are clear in that there is more attack signal over promoter/enhancer regions of FOXA2 in cells that express FOXA2. ATAC chromatin accessibility is associated with gene expression so this is to be expected and does add much if anything to the mechanistic understanding of how ASCL1 is regulating FOXA2. What comes first, ASCL1+cofactors to open the site, or some independent factors opening these sites and increase the probability of ASCL1 binding. These arguments become circular. The data are consistent with ASCL1 being a player in FOXA2 expression but not sufficient. Just showing different ATAC does not provide the basis for how. Are there motifs in the ATAC peak regions for FOXA2 that suggest cofactors? This could be discussed if any.

5) The IF panels Fig. 6a and extended Fig. 6a show ASCL1 protein with lower expression in the never-met primary tissues

(n=3) compared to the met-associated primary tissues (n=3) that express FOXA2. What are the relative RNA levels in these samples and what is the ATAC signal at the ASCL1 locus? Is there a levels argument to make with either RNA or protein? Might ASCL1 protein stability be a player?

6) Proliferation assay description is insufficient in methods.

7) The RNA-seq and ATAC-seq were not provided at the time of review. Useful data sets were generated that will be valuable for the field. It is assumed they will be publicly available at publication.

Reviewer #3

(Remarks to the Author)

Summary

The study by Kawasaki et al. investigates a critical feature of small cell lung cancer (SCLC), characterized by their exceptionally high metastatic potential, with two-thirds of patients presenting with distant metastases at the time of diagnosis, resulting in a poor prognosis. The authors propose that genetic alterations underlie cancer initiation, while non-genetic modifications play a central role in metastasis initiation. They identify ASCL1 and FOXA2 as pivotal drivers of mutagenesis and metastasis formation in SCLC, respectively.

While the interplay between ASCL1 and FOXA2 is not a novel finding, this regulation has not previously been demonstrated in SCLC. The study's identification of FOXA2 as the primary driver of metastasis and a direct regulator of ASCL expression represents a key contribution, with potential implications for developing targeted therapies for SCLC. I am inclined to recommend the manuscript for publication, as it presents a compelling and well-constructed narrative. However, there are several concerns that must be addressed prior to publication.

Major Concerns

1. Bioinformatic Analysis Transparency

The manuscript does not provide sufficient detail regarding the bioinformatic methods, and no associated code or pre-processed data has been made available for validation. To address this, I strongly recommend the authors:

- o Upload all analysis code to a public repository (e.g., GitHub) and include the repository link in the manuscript.
- o Make pre-processed single-cell data (e.g., .h5ad or .Rds formats) and ATAC-seq data (e.g., .bigwig files) accessible via public platforms such as GEO or Zenodo, accompanied by dataset IDs, DOIs, and links.

2. FFPE-ATAC Experiment

While FFPE-ATAC is technically impressive, there are unresolved concerns regarding cell viability in these tissues. Cell viability is critical for chromatin accessibility assays, as dying/dead cells can introduce significant artifacts. The authors should:

- o Provide cell viability data for each sample in a table.
- o Clarify whether selection of viable cells was performed prior to DNA isolation.
- o If the entire FFPE block was used without enrichment for tumor cells, explain how the analysis accounts for non-tumor cells potentially confounding the results.
- o If no selection was performed, conduct in silico cell type decomposition to enrich tumor-specific signals.

3. Hypothesis Validation

The manuscript relies on chromatin accessibility changes to support its hypothesis, which is insufficient to directly establish causation. To strengthen the conclusions:

- o Perform additional experiments to disrupt ASCL1 binding at FOXA2 regulatory elements. For example, CRISPR-mediated deletion of ASCL1 binding motifs or the regulatory elements themselves could demonstrate the functional necessity of this interaction.
- o Evaluate the metastatic potential of cells with disrupted ASCL1-FOXA2 interactions in an in vivo model, such as mouse engraftment assay. A reduction in metastasis compared to controls would directly support the authors' claims.

4. FOXA1-FOXA2 Compensation

Given the compensatory roles of FOXA1 and FOXA2 (as previously reported in <https://doi.org/10.1074/jbc.M414122200> and others), the authors should assess whether FOXA1 compensates for FOXA2 loss. Specifically:

- o Examine changes in FOXA1 chromatin accessibility after FOXA2 deletion.
- o Determine whether FOXA1 functionally substitutes for FOXA2 in driving metastasis, as this would have important therapeutic implications.

Minor Concerns

1. scRNA-seq Analysis

- o The sample groups are unbalanced, with relatively few cases lacking metastasis at diagnosis. While this limitation is understandable, the authors should refrain from making overly bold claims based on the small sample size.
- o The identification of epithelial cells as tumor cells requires additional validation. InferCNV alone is insufficient to confirm tumor identity. The authors should use complementary approaches, such as FISH analysis for known chromosomal aberrations, to confirm tumor cell populations.

o Potential confounding by healthy epithelial cells

Based on publicly available single-cell atlases of healthy lungs (e.g., linked datasets below), ACSL1 and FOXA1 are highly co-expressed in alveolar type II epithelial cells. This raises the possibility that observed differences may reflect variations in the proportion of cancer versus healthy epithelial cells. The authors should account for this by explicitly quantifying cancer epithelial cell contributions.

<https://cellxgene.cziscience.com/e/9f222629-9e39-47d0-b83f-e08d610c7479.cxg/>

<https://cellxgene.cziscience.com/e/066943a2-fdac-4b29-b348-40cede398e4e.cxg/>

2. PCA Analysis (Extended Fig. 2D)

Principal component analysis (PCA) suggests the presence of technical or other confounding factors. The authors should:

- o Provide additional PCA plots (e.g., PC2/PC3, PC3/PC4) to better evaluate batch effects or technical artifacts.
- o Include a heatmap showing primary non-metastatic, primary metastatic, and metastasis samples in the same plot for better assessment of the variability.

3. Data Visualization

o Fig. 2E: Replace the current visualization with violin plots, showing each sample as a distinct data point, to improve interpretability.

o Fig. 4G-J: Clarify whether "set size" refers to the number of differentially expressed (DE) genes in a pathway or the total number of genes in the pathway database. Ideally, display the number of DE genes per pathway to avoid misinterpretation.

o Fig. 5B: The integration of single-cell RNA-seq data appears flawed or incomplete. Without code or a detailed methods description, it is difficult to assess this issue.

4. Experimental Design and Comparisons

o The comparison of "never-met primary vs. met-associated metastasis" is unconventional. A more logical design would compare "never-met primary vs. met-associated primary" and "met-associated primary vs. met-associated metastasis" (ideally matched samples). The authors should justify their chosen comparisons or revise their analysis.

5. Other Methodological Concerns

o GSEA Analysis: Genes are typically ranked by logFC rather than p-values. The authors should clarify their rationale for using p-values and specify whether pre-filtering based on logFC was performed, prior to gene ranking.

o ChIP-seq/qPCR: Input DNA amounts are highly variable (0.02–5 ng), which can introduce bias. For cell line experiments, such low input values are unjustifiable. The authors should provide additional justification or repeat these experiments with consistent, sufficient input amounts.

o Fig. 6C, E: The data quality is unclear. The authors should:

Include positive control regions (constitutively open/accessibly chromatin) to demonstrate data reliability.

Add scale bars to plots for accurate interpretation of chromatin accessibility.

6. Prospective Implications

o Could metastasis be predicted in a prospective study based on ASCL1 and FOXA2 expression at diagnosis?

o Would longer follow-up alter conclusions regarding "never-met" patients with slower-growing tumors? These points should be discussed to contextualize the findings.

Conclusion

This manuscript offers valuable insights into the mechanisms underlying SCLC metastasis. However, several issues related to methodology, experimental validation, and data interpretation must be addressed. By providing greater transparency, conducting additional functional experiments, and refining data analyses, the authors can significantly strengthen their study.

Reviewer #4

(Remarks to the Author)

This manuscript investigates the role of FOXA2 in driving metastasis in small cell lung cancer (SCLC). The authors leverage a large clinical cohort, in vivo models, and multiple omics techniques to demonstrate a strong association between FOXA2 expression and metastatic capacity. Further investigation using in vivo models demonstrated that FOXA2 promotes multi-site metastasis and induces a fetal neuroendocrine gene expression program. The authors identify a critical regulatory mechanism: ASCL1, a known promoter of SCLC tumorigenesis, directly binds to the FOXA2 promoter and regulates its expression. This ASCL1-FOXA2 axis is proposed as a novel driver of multiorgan SCLC metastasis. Clinically, high FOXA2 expression in primary SCLC is identified as a potential predictive biomarker for relapse. Finally, in vivo experiments demonstrate that FOXA2 knockdown significantly reduces metastasis without impacting tumor growth.

While the study presents compelling data linking FOXA2 expression to metastatic potential, several critical aspects require closer examination before publication can be recommended.

(1) The mechanistic details connecting FOXA2 to the observed fetal neuroendocrine gene expression program and subsequent metastasis remain relatively superficial. The connection between FOXA2, the fetal neuroendocrine gene expression program, and the specific processes involved in metastasis (e.g., EMT, cell migration, invasion, extravasation)

remains unclear. The current evidence is primarily correlative. Further investigation into the downstream pathways activated by FOXA2 and its interaction with other metastasis-related genes is needed.

(2) Although in vivo metastasis assays are performed, the manuscript lacks further functional assays to directly test the impact of FOXA2 on specific aspects of metastasis (e.g., cell migration, invasion, adhesion). Such assays would strengthen the mechanistic understanding.

(3) The manuscript identifies ASCL1 as a regulator of FOXA2, but the precise molecular mechanisms through which ASCL1 activates FOXA2 remain unclear. Are all ASCL1 high tumors have high FOXA2 levels? Are there ASCL1 high tumors that show low expression of FOXA2? Does FOXA2 have different effects in different cell populations within the tumor? Are there specific enhancer regions involved in FOXA2 regulation? What other transcription factors or epigenetic modifications are at play? The study lacks the depth of mechanistic analysis needed to truly understand the regulatory network surrounding FOXA2.

(4) The majority of the in vivo experiments utilize only two SCLC cell lines (H1836 and SHP-77). In addition, SHP77 cell line expresses untypical for SCLC oncogenic KRAS(G12V) mutant and lacks of characteristic for SCLC RB1 mutations. It is crucial to replicate the findings in additional cell lines and, ideally, in a wider range of PDXs to confirm the generalizability of the findings beyond this limited set. In addition, ASCL1-low and ASCL1-high cell lines should be used to validate the proposed ASCL1-FOXA2 signaling axis.

(5) The authors acknowledge that the never-metastatic cohort might not represent truly non-metastatic disease. Some tumors in this group might represent early-stage disease caught before widespread metastasis. This ambiguity significantly weakens the comparison and raises concerns about the generalizability of the findings. A more rigorous definition, perhaps incorporating tumor stage and survival time beyond a simple 2-year threshold, along with additional biological characterization of the "never-metastatic" tumors is necessary.

Version 1:

Reviewer comments:

Reviewer #1

(Remarks to the Author)

I would like to acknowledge the high-quality work performed by the authors in addressing my comments and complementary comments from 3 other reviewers.

I have no further concerns.

Reviewer #2

(Remarks to the Author)

The authors have carefully and thoughtfully addressed all my comments and concerns. These revisions and clarifications have improved the manuscript in my opinion. I have no further concerns.

Reviewer #3

(Remarks to the Author)

I appreciate the authors thorough revision, which have adequately addressed all my concerns and questions. I have no further comments and recommend the manuscript for publication in Nature Communications.

Reviewer #4

(Remarks to the Author)

The author's consideration and response to the earlier remarks is appreciated. however, the response falls short in critical points. A few remaining issues are:

-It is unclear why FOXA2 depletion was not technically possible in any additional cell lines, given that the Authors claim FOXA2 does not affect normal cell growth. Therefore, it is likely that FOXA2 is a critical survival factor in those cells, undermining the Author's claim that FOXA2 is predominantly a regulator of metastasis.

-In addition, the Authors write (in response to R2's important question regarding the mechanisms of ASCL1 binding to FOXA2) that: single-cell sorting after CRISPR targeting is not possible in H1836 cells. That further supports the need to perform the experiments in additional cell lines in which mechanistic experiments will be possible.

-Requested direct testing of the impact of FOXA2 on specific aspects of metastasis like cell migration or invasion assays, was not performed. Instead, the Authors performed colony formation assays, which showed that FOXA2 depletion robustly

(up to ~ 90%) inhibits the tumor-forming properties of SCLC cell lines. That is in contradiction with the xenograft assays, which indicate that there is no defect in tumor-forming properties.

-Of note, further description in the Methods section of colony formation assays would be necessary as it is not obvious how this experiment was performed with suspension cells.

-No further mechanistic insight on how FOXA2 regulates target genes is presented in the revision. Chromatin accessibility changes are insufficient to directly establish causation. Given that the previous publications (PMID: 37963187) established the critical role of FOXA1/2 in SCLC metastasis, the presented manuscript is somewhat more incremental.

Version 2:

Reviewer comments:

Reviewer #4

(Remarks to the Author)

I appreciate the authors' candid and honest response. I fully acknowledge that SCLC lines can be temperamental during viral transductions and in the selection process. I kindly suggest that the authors include a brief overview of the negative results observed in knockdown experiments conducted in other cell lines. I believe the editor will concur that presenting these results will not detract from the manuscript; rather, it will inform readers that attempts were made to reproduce findings in suspension SCLC lines, which are more commonly used for SCLC growth studies. Including this information would enhance the manuscript by providing a more comprehensive understanding of the challenges associated with these experimental approaches in SCLC.

RESPONSE TO REVIEWERS

We thank the reviewers and for their several thoughtful suggestions regarding this work. We have addressed each of the critiques raised by all four reviewers point by point in the response below. We believe that incorporating their input and clarifying some areas has substantially strengthened this paper. Comments are reproduced verbatim, followed by responses in blue. Text changes are indicated in red.

Reviewer #1

1. While the experimental design is original and results are robust, the novelty of the findings is not obvious as ASCL1-FOXA2 has previously been linked to drive SCLC metastasis (<https://doi.org/10.1158/0008-5472.CAN-23-1079>). Furthermore, the authors note that ASCL1 expression is far more prevalent than FOXA2, and SCLC exists with or without co-expression of FOXA2. This implies that ASCL1 likely drives a broader transcriptional program beyond FOXA2. Characterizing this larger ASCL1-driven program could be more impactful than focusing on FOXA2 alone. Additionally, ASCL1-high/FOXA2-low cases exist, suggesting that there are additional, unidentified regulators of FOXA2. These factors should be explored to complete the story.

The reviewer is quite correct that both ASCL1 and FOXA2 have been previously implicated in SCLC development and, to a limited extent, metastasis, in the Ko et al. paper referenced above. However, there are key novel points in this manuscript including:

1. Our experimental data are all from human SCLC while Ko et al focuses on mouse models of the human disease. While highly valuable, mouse models do have critical differences from the human disease, including rarity of brain metastases, absence of tobacco mutational signature, low tumor mutation burden, relative genomic stability, and NFIB as a driver of metastasis.
2. The main focus of the Ko et al paper is on a distinction between NFIB-driven and NFIB-independent pathways of metastasis: the data on ASCL1 and FOXA1/2 is limited. The authors note that FOXA1 and 2 are two of the candidate regulators indentified by ATAC-seq and that manipulation of these genes will be required in further studies. These data are provided for the first time in our manuscript, and in the human disease being modeled.

With regard to ASCL1 having a broader role, the reviewer is exactly correct. ASCL1 is a master transcriptional regulator of neuroendocrine cell fate determination in multiple developmental lineages and a primary driver of SCLC oncogenesis. A broad-based analysis of the downstream pathways of ASCL1 would certainly be an interesting paper – but one addressing a different question entirely. The key observation in this paper is that a particular signaling axis, ASCL1 to FOXA2, is specifically implicated as a promoter of metastasis in human SCLC. This observation begins with the identification of FOXA2 as the lead candidate associated with metastasis in a human tumor dataset, followed by confirmation through *in vivo* models. While FOXA2 is further characterized here by ChIP-Seq as a downstream target of ASCL1, ASCL1 absolutely has distinct and pleotropic roles, including that exemplified in our paper by the ASCL1-PROX1 signaling axis which is independent of FOXA2, and not implicated in metastasis.

We added to the discussion, referencing the Ko et al. paper and emphasizing both the consistency with these results and the novelty and potential impact of the current work.

Line 500: Recent observations in a genetically engineered mouse model of SCLC nominated FOXA1 and/or FOXA2 as candidate factors associated with SCLC metastasis [Ko J et al, Cancer Res 2024]. The identification and characterization of FOXA2 as a contributor to metastatic competency in primary human SCLC is consistent with the predictions of the mouse model. We also note some key differences in apparent drivers of metastasis in human and murine SCLC – perhaps most notably NFIB, which is strongly associated with SCLC metastasis in the mouse; we have not observed differentially elevated *NFIB* expression in tumors with metastatic potential in our clinical datasets. FOXA1 levels in SCLC models were unaltered by either FOXA2 KD using the 1st hairpin, or in reponse to FOXA2 overexpression (data not shown), arguing against a compensatory role for FOXA1 and emphasizing the importance of FOXA2 for the observed effects. Our data support FOXA2 as a candidate predictor of SCLC relapse in patients with definitively treated early stage disease.

The reviewer is also correct that other factors beyond ASCL1 must affect FOXA2: while ASCL1 does bind to and increase expression of *FOXA2*, the two genes do not vary in lockstep. Fully characterizing the many potential protein complexes affecting *FOXA2* gene expression is beyond the scope of the current manuscript. These transcriptional regulatory complexes may include dynamic interactions of both positive and negative factors. For

the reviewer's interest, in the figure below we illustrate an approach we have considered to defining candidate positive co-regulators. In our datasets, there are 765 significantly differentially expressed genes upregulated and associated with metastasis; and 258 genes demonstrating significant co-expression with FOXA2. Analysis of the FOXA2 gene locus for peaks from ASCL1-ChIPseq and ATACseq suggests 226 candidate factors that might bind to their putative consensus sites in this locus. Cross-referencing these datasets, if we hypothesize that the most relevant factors might be those with consensus binding motifs found in FOXA2 and not in PROX1 (as a representative ASCL1-dependent gene NOT associated with metastasis), this defines 26 candidates. 15 candidate genes are highly expressed in metastasis and correlate with FOXA2 (but are not among those implicated as possible direct binders of the FOXA2 gene locus); 7 among the 226 potential binders are FOXA2 vs. PROX1-unique (but not upregulated in metastasis); and 6 among the 226 are differentially expressed in metastasis and significantly correlated with FOXA2. One factor (SOX2) is identified as a potential binder to enriched motifs of the FOXA2 gene and is co-expressed with FOXA2, but not upregulated in metastasis. An equally complex set of possible negative regulators could be considered. Given this complexity, we would propose future studies of the regulation of FOXA2 expression starting with a mutational scan across the FOXA2 upstream promoter/ enhancer region in SCLC lines to define active binding elements affecting gene expression beyond ASCL1.

[REDACTED]

Please note we have not included these data in the revised manuscript, as we think this may detract from the focus on the ASCL1-FOXA2 axis, but we would be willing to include as a supplementary panel if the editor requests. We have added a caveat to the analysis to recognize this issue, as follows:

Line 430; Given that not all ASCL1-positive SCLC express FOXA2, additional positive and/or negative regulators are likely to be involved in controlling FOXA2 expression. Identification and characterization of these additional co-factors through detailed interrogation of ASCL1 co-factors or post-translational modifications, and of FOXA2 gene regulatory elements and chromatin status, will be the focus of future analysis.

2. ASCL1-FOXA2 Axis Validation:

- a. Does ASCL1 knockdown abolish FOXA2 expression and metastatic propensity in metastatic SCLC models? The authors show in Fig 5h, that indeed knockdown of ASCL1 decreased FOXA2 protein levels in H1836 and SHP-77 cell lines, the effect on metastatic propensity seems to be not included. The authors should include these results.

As the reviewer notes, we have shown that knockdown of ASCL1 decreases FOXA2. ASCL1 knockdown would indeed decrease metastasis – but the experiment cannot be interpreted as a specific effect on metastasis because, as noted above, ASCL1 is also a critical driver of SCLC oncogenesis. Unlike FOXA2 knockdown, ASCL1 knockdown in ASCL1-expressing human SCLC inhibits primary clonogenic and tumor-initiating capacity (Jiang et al., Cancer Res 2009), and in genetically engineered mouse models, Ascl1 knockout prevents SCLC tumor development (Boromeo et al, Cell Rep 2016). Thus a specific effect on metastasis cannot be assessed.

- b. Does the ASCL1-FOXA2 regulatory axis only apply to ASCL1-positive SCLC subtypes? Would FOXA2 KD in none ASCL-1 subtypes have no impact on metastasis?
- c. Are all FOXA2-high tumors also ASCL1-high?

While the ASCL1-FOXA2 axis defined here would be relevant only to ASCL1-expressing subtypes, our original RNA-seq data notably includes 2 samples which do not express detectable ASCL1 but are FOXA2-positive. Both also express YAP1, and both are from metastatic sites (brain and bone). We do not have matched primary samples from these cases: the diagnostic biopsies were from the metastatic sites. Thus we are unable to characterize potential subtype heterogeneity in the primary tumor (i.e. could these have started as ASCL1+?), an important caveat given the heterogeneity and the plasticity of the cell states which has been reported (Chan et al., Cancer Cell 2021, Duplaquet et al., Nat Cell Biol 2023, Redin et al., J Hematol Oncol 2024, Ireland et al., Cancer Cell 2020). It is possible that these evolved from ASCL1-expressing precursors, but their presence does imply that there must be ASCL1-independent mechanisms for activating FOXA2 expression. This has been also shown in Extended data Fig5b.

Querying all SCLC cell lines in the CellMiner database, there are no available ASCL1-low/FOXA2-high lines in

which to test the idea of FOXA2 KD in a non-ASCL1-expressing SCLC line: all available YAP1-high and POU2F3-high lines are FOXA2-low (see co-expression plots below). In response to the reviewer suggestion, we have added the following to the manuscript.

Line 486; It should be additionally emphasized that the ASCL1-FOXA2 axis is of relevance for NE-high subtypes of SCLC that express ASCL1. Other determinants of metastasis are likely to be of primary relevance in NE-low subtypes of SCLC.

3. To further functionally validate the ASCL1-FOXA2 Axis:

- a. The authors should explore if (low-moderate) overexpression of FOXA2 and/or ASCL1 in non-metastatic SCLC is sufficient to drive metastasis. This would provide stronger evidence for the role of the ASCL1-FOXA2 Axis as drivers of SCLC metastasis.
- b. Specifically, does overexpression ASCL1-FOXA2 Axis lead to increased metastatic potential or penetrance, and does it preferentially promote liver metastasis? This is particularly relevant since shFOXA2 greatly suppresses liver metastases (Extended Data Fig. 3e), while its effects on adrenal and ovarian metastases are less pronounced. This hints at potential organotropism, mediated by FOXA2 which would be conceptually very interesting.

We considered two distinct scenarios in addressing these questions – first, in FOXA2-high SCLC, is FOXA2 promoting metastasis; second, in FOXA2-low SCLC, which may use alternative mechanisms to drive metastasis, could FOXA2 further augment this. In our original submission, we knocked down FOXA2 in two FOXA2-high lines, demonstrating that this KD led to a marked suppression of metastasis in both. To further demonstrate specificity of this effect, we now include a critical add-back experiment. Human and murine FOXA2 demonstrate 97% homology, but one of the FOXA2 shRNA used was human specific, allowing us to restore FOXA2 activity in the KD line with the mouse FOXA2 cDNA. This shows clear restoration of metastatic potential in FOXA2 KD model suggesting that FOXA2 is indeed required in this line for high metastatic competency.

To address the second scenario, we noted that following intracardiac injection of the H1963 line (ASCL1+ but FOXA2-low), metastases were infrequent (1 out of 6, and 1 out of 8 mice, both ovarian metastases). The ovarian metastases demonstrated upregulation of FOXA2 relative to the parental line, suggesting that subclones

expressing higher levels of FOXA2 might be selectively contributing to metastasis (see Figure below). However, exogenous expression of FOXA2 in parental H1963 cells did not appreciably alter the frequency or distribution of subsequent metastases – intracardiac injection of this clone also demonstrated metastasis only to ovary, and at the same frequency as control.

Taken together, these data suggest that (1) in ASCL1/FOXA2-positive SCLC, FOXA2 is necessary for high-frequency metastasis, but (2) FOXA2 is not sufficient to augment metastatic frequency of FOXA2-low SCLC, in which metastatic competency may be determined by alternative mechanisms. We have summarized these new data in the revised manuscript as follows:

Line 292; To further demonstrate specificity of metastatic inhibition by FOXA2 KD, we performed an add-back experiment in the SHP-77 KD line. Human and murine FOXA2 demonstrate 97% homology [Jackson BC et al. Hum Genomics, 2010]. Despite this close homology, one of the FOXA2 shRNA used was human specific, allowing us to restore FOXA2 activity in KD cells with murine FOXA2 cDNA. Restoration of FOXA2 expression was confirmed by western blot (**Extended Data Fig.3g**). Intra-cardiac injection of FOXA2-restored cells in mice showed clear re-establishment of metastatic potential in FOXA2 KD cells (**Fig.3g,h**), confirming that FOXA2 is indeed required in this line for high metastatic competency.

Next we sought to investigate the effects of exogenous FOXA2 expression in a FOXA2-low cell line, H1963. Intracardiac injection of H1963 resulted in rare metastases, exclusively to the ovary (1 out of 6, and 1 out of 8 mice, in two independent experiments). The ovarian metastases demonstrated upregulation of FOXA2 by immunohistochemistry relative to the subcutaneous tumor generated from the parental line (**Fig.3i**), implying that subclones expressing FOXA2 might be selectively contributing to metastasis. An H1963 subline generated from an ovarian metastasis (**Fig. 3j**) indeed showed persistent strong enrichment of FOXA2 relative to the parental line (**Fig.3k**). However, intracardiac injection of H1963 cells modified to exogenously express FOXA2 did not appreciably alter the frequency or distribution of subsequent metastases: this clone also demonstrated metastasis only to ovary, and at the same frequency as control (data not shown). Thus while FOXA2 is strongly associated with metastasis and appears to be required for high metastatic competency in FOXA2-high SCLC, other factors may be more important determinants of metastasis in FOXA2-low contexts.

4. What is the organotrophic profile of metastases in FOXA2-high patients? Is there a liver preference in these cases, consistent with preclinical findings?

This is an interesting suggestion, but the data we have does not convincingly demonstrate any specific organotrophic shifts associated with high FOXA2 expression. The graph below is derived from RNA-seq data from the metastatic lesions of 25 patients in the initial cohort. Liver metastases are common, but do not appear to be selectively enriched for high FOXA2 relative to other metastatic sites. We have noted this in the revised manuscript, as follows:

Line 287; The predominance of liver metastases from these FOXA2-high cell lines raised the question of whether FOXA2, in addition to promoting metastasis, might alter organ tropism toward liver metastasis. We further explored the distribution of FOXA2 levels among 25 metastatic sites in our patient cohort. Liver metastases do not appear to be enriched among high FOXA2 tumors relative to other metastatic sites (**Extended Data Fig.3f**).

5. Were there any differences in TMB and could this influence metastatic potential between never-metastatic and metastatic SCLC cases. Might be a point to discuss or address as it should be available within the current datasets.

We investigated TMB in these cohorts using MSK-IMPACT. There were no differences in these 2 groups by comparing never-mets vs metastasis to non-synonymous mutation (Mann-Whitney U test). We have noted this in the revised manuscript:

Line 187; No consistent mutational differences between cohorts were observed including tumor mutation burden (Extended Data Fig.2e).

Minor Comments:

1. Many graphs are missing y-axis ticks, making interpretation difficult.

Thank you for pointing this out. We added the y-axis ticks in all the figure panels for clarity.

2. Figures 3 and 4 could be combined into a single figure for clarity and better coherence with the rest of the figures. Overall the figures would do well with a uniform standardization.

With the addition of the mouse FOXA2 addback data and the intra-cardiac injection of H1963, we extended Figure 3 and Extended Figure 3, which now makes it difficult to merge with Figure 4.

3. Figure legends would benefit from declarative titles that summarize key findings (e.g., "FOXA2 Knockdown Reduces Liver Metastases").

We have modified to include declarative figure legend titles where appropriate – although for Figure 3, considering the additional data on organ tropism, we did not specify liver metastasis.

Reviewer #2

1) The knock down of FOXA2 showing loss of metastatic efficiency in 2 SCLC cell line models is the important experiment shown supporting one of the main conclusions in the study, that being the requirement for FOXA2 for metastasis in SCLC. From what is shown, the results are clean and dramatic. However, the data should be shown more thoroughly. For example, data points for the individual mice should be shown on the graphs in Fig. 3c-f. Some of the extended Fig. 3 data (such as extended Fig. 3c-e) could be moved into the main figure due to the importance of these experiments. Additional experiments

shown in Fig. 4a-f showing FOXA2 is not required for xenograft tumor growth also need individual animals shown.

We agree, and have revised to include each data point in Fig 3c-f, Extended data Fig3d,e, and Fig4 a-f.

2) As stated above, the most compelling finding is the apparent requirement for FOXA2 in the in vivo metastasis test using SCLC cell lines introduced intracardially in mice. An important complement to these findings is testing a SCLC cell line with low FOXA2 normally but is engineered to exogenously express FOXA2-- tested for liver metastasis as per the assay used here. The cells have already been developed and used in RNA-seq experiments in his study. Another experiment that would significantly strengthen the arguments for the role of FOXA2 in metastasis is manipulating levels of FOXA2 in the PDX FOXA2 high and low models and testing occurrence of mets.

We appreciate these suggestions. As outlined in response to **Reviewer 1 Point 3**, we have now performed add-back experiments in the KD lines to demonstrate specificity of the effect on metastasis, and exogenous expression in the ASCL1+/FOXA2-low line H1963; please see summary of results above. In addition, as the reviewer suggested, we tried to derive tumoroids from five of our SCLC PDX models. Unfortunately, only one of these remained viable until day 43, and this one could not be passaged.

3) What happens to ASCL1 levels with FOXA2 KD? Is there a feedback relationship? FOXA2 as a pioneer factor may increase ASCL1 levels consistent with the IFs of tissue samples shown in Fig. 6.

While ASCL1 controls FOXA2 levels, the converse does not appear to be the case. We assessed expression of ASCL1 in H1836 and SHP-77 transduced with shRNA to FOXA2, and see no consistent effect on ASCL1 levels. We have emphasized this in our revised manuscript.

Line 389; FOXA2 KD did not demonstrate a consistent effect on expression of either ASCL1 or PROX1 (Extended Data Fig.5c).

4) The ATAC data are clear in that there is more attack signal over promoter/enhancer regions of FOXA2 in cells that express FOXA2. ATAC chromatin accessibility is associated with gene expression so this is to be expected and does add much if anything to the mechanistic understanding of how ASCL1 is regulating FOXA2. What comes first, ASCL1+cofactors to open the site, or some independent factors opening these sites and increase the probability of ASCL1 binding. These arguments become circular. The data are consistent with ASCL1 being a player in FOXA2 expression but not sufficient. Just showing different ATAC does not provide the basis for how. Are there motifs in the ATAC peak regions for FOXA2 that suggest cofactors? This could be discussed if any.

The reviewer is entirely correct that ATAC-seq data do not provide specific information about what determines ASCL1 binding to the FOXA2 locus. However, informative insights are rather provided by ASCL1 ChIP-seq and ASCL1 ChIP-qPCR assays. ATAC-seq does indicate areas of open chromatin, that could be further interrogated for other binding factors influencing FOXA2 expression positively or negatively. As described in response to **Reviewer 1 Point 1**, we have begun to map out possible candidates through a combination of analyzing consensus binding motifs found in the FOXA2 locus, factors co-expressed with FOXA2, and factors enriched in SCLC metastasis. Further detailed analyses of positive and negative regulation of FOXA2 expression would require a mutational scan across the FOXA2 upstream promoter/ enhancer region in SCLC lines to define active binding elements affecting gene expression beyond ASCL1, followed by identification of putative binding factors and detailed characterization of their activities alone and in combination. We have acknowledged this as a future direction in the revised manuscript:

Line 430; Given that not all ASCL1-positive SCLC express FOXA2, additional positive and/or negative regulators are likely to be involved in controlling FOXA2 expression. Identification and characterization of these additional co-factors through detailed interrogation of ASCL1 co-factors or post-translational modifications, and of FOXA2 gene regulatory elements and chromatin status, will be the focus of future analysis.

5) The IF panels Fig. 6a and extended Fig. 6a show ASCL1 protein with lower expression in the never-met primary tissues (n=3) compared to the met-associated primary tissues (n=3) that express FOXA2. What are the relative RNA levels in these samples and what is the ATAC signal at the ASCL1 locus? Is there a levels argument to make with either RNA or protein? Might ASCL1 protein stability be a player?

Material is not available from these small formalin-fixed samples for RNA-seq. However, to further support the association of metastasis with ASCL1-associated expression of FOXA2 specifically, we added the IF staining of PROX1, as a second factor downstream of ASCL1. These data show ASCL1 and PROX1 expression in never-mets but not FOXA2 expression. This result supports the expression data from the initial cohort, in which ASCL1, PROX1, CACNA1A and SOX2 did not show consistent differences between never-met and met cohorts, while FOXA2 did. We have included these data as Fig6a and Extended Fig6a.

6) Proliferation assay description is insufficient in methods.

We apologize for the oversight and have added a description of the method used:

Line 726; Proliferation assay

Cell proliferation assays were performed with 1000 cells/well (SHP-77) and 2000 cells/well (H1836) seeded in 96 well-plates. The cells were treated with doxycycline starting on day 4 at 1 µg/mL and replenished every 48 hr through day 10. Luminescence was measured using the CellTiter-Glo 2.0 Assay (Promega, G9242) following the manufacturer's instructions. The proliferation rate was calculated by normalizing to day 1.

7) The RNA-seq and ATAC-seq were not provided at the time of review. Useful data sets were generated that will be valuable for the field. It is assumed they will be publicly available at publication.

Absolutely. We have now deposited these data and we added the GEO accession numbers to the manuscript; GSE281523, GSE281524, GSE281525, GSE281740

Reviewer #3

Major Concerns

1. Bioinformatic Analysis Transparency

The manuscript does not provide sufficient detail regarding the bioinformatic methods, and no associated code or pre-processed data has been made available for validation. To address this, I strongly recommend the authors:

- Upload all analysis code to a public repository (e.g., GitHub) and include the repository link in the manuscript.

- Make pre-processed single-cell data (e.g., .h5ad or .Rds formats) and ATAC-seq data (e.g., .bigwig files) accessible via public platforms such as GEO or Zenodo, accompanied by dataset IDs, DOIs, and links.

Thank you for this comment. As noted above, we have deposited these data and we added the GEO accession numbers to the manuscript; GSE281523, GSE281524, GSE281525, GSE281740

We also uploaded our code in the following URL and included in our manuscript; https://github.com/shahcompbio/SCLC_MET.git

2. FFPE-ATAC Experiment

While FFPE-ATAC is technically impressive, there are unresolved concerns regarding cell viability in these tissues. Cell viability is critical for chromatin accessibility assays, as dying/dead cells can introduce significant artifacts. The authors should:

- Provide cell viability data for each sample in a table.
- Clarify whether selection of viable cells was performed prior to DNA isolation.
- If the entire FFPE block was used without enrichment for tumor cells, explain how the analysis accounts for non-tumor cells potentially confounding the results.
- If no selection was performed, conduct *in silico* cell type decomposition to enrich tumor-specific signals.

We appreciate these concerns. H&E-stained sections of tumors used for FFPE-ATAC were reviewed by a thoracic pathologist to ensure that the areas profiled in all cases had high viability (>90%), and were not necrotic. Viability of FFPE-derived cells was also examined and it is now included in (Extended Fig.6b). Of particular importance for this assay is 3D intact nuclei. Below, we show the DAPI staining of each FFPE sample where most cells show 3D intact nuclei as indicated by smooth perinuclear staining; >92% of nuclei in each sample were intact. Selective enrichment for tumor cells is not possible from FFPE material, but SCLC tumors characteristically manifest tightly packed sheets of cancer cells with minimal stromal content. Microdissection was performed to select areas of the slide with high intact tumor content. Finally, our question was focused on differential chromatin accessibility in FOXA2- (never metastasized) and FOXA2+ (metastasis-associated primary) cases. We would consider the confounding from non-tumor contamination will be very limited in this context. We have noted in the manuscript;

Line 409; The tumors processed for FFPE-ATAC analysis showed predominantly 3D intact nuclei as defined by DAPI staining (**Extended Data Fig.6b**).

3. Hypothesis Validation

The manuscript relies on chromatin accessibility changes to support its hypothesis, which is insufficient to directly establish causation. To strengthen the conclusions:

- Perform additional experiments to disrupt ASCL1 binding at FOXA2 regulatory elements. For example, CRISPR-mediated deletion of ASCL1 binding motifs or the regulatory elements themselves could demonstrate the functional necessity of this interaction.
- Evaluate the metastatic potential of cells with disrupted ASCL1-FOXA2 interactions in an *in vivo* model, such as mouse engraftment assay. A reduction in metastasis compared to controls would directly support the authors' claims.

We agree with the reviewer that the chromatin accessibility changes are insufficient to directly establish causation. The primary claim of this paper is that FOXA2 promotes SCLC metastasis, based on expression analysis of primary human SCLC and validation by manipulation of expression in *in vivo* models, and the chromatin accessibility data are supportive.

We are unable to perform the suggested targeted gene editing of the FOXA2 regulatory elements for technical

reasons – this would require single cell sorting after CRISPR targeting and the SCLC lines used here are suspension cells that grow in clusters and do not recover from single cell sorting. This becomes even more problematic in targeting a locus likely to result in FOXA2 suppression: in FOXA2-expressing SCLC, we have shown that FOXA2 KD cells recover poorly both in vitro and in vivo (colony formation assay, metastatic assay). For example, seeding 96 cells of the parental H1836 and of the two H1836 FOXA2 KD lines in 96 well plate resulted in no viable recovery. However, the absence of such experiments does not compromise the validity of our primary claims that FOXA2 promotes SCLC metastasis and, further, that differences in ASCL1 access to the FOXA2 locus are associated with FOXA2 expression.

4. FOXA1-FOXA2 Compensation

Given the compensatory roles of FOXA1 and FOXA2 (as previously reported in <https://doi.org/10.1074/jbc.M414122200> and others), the authors should assess whether FOXA1 compensates for FOXA2 loss. Specifically:

- Examine changes in FOXA1 chromatin accessibility after FOXA2 deletion.
- Determine whether FOXA1 functionally substitutes for FOXA2 in driving metastasis, as this would have important therapeutic implications.

Our data does strongly point to a FOXA2-specific phenotype. The 1st hairpin used for FOXA2 KD in H1836 and SHP-77 specifically suppresses FOXA2, with no effect on FOXA1, and the FOXA2 overexpression construct used in H1963 cells upregulates FOXA2 with no evident compensatory change in FOXA1 (Figure below). The 2nd hairpin used would target both FOXA1 and 2 due to homology, but similar effects in vitro and in vivo are seen with both hairpins used. We noted this in the discussion as follows;

Line 506; FOXA1 levels in SCLC models were unaltered by either FOXA2 KD using the 1st hairpin, or in response to FOXA2 overexpression (data not shown), arguing against a compensatory role for FOXA1 and emphasizing the importance of FOXA2 for the observed effects.

[REDACTED]

Minor Concerns

1. scRNA-seq Analysis

- The sample groups are unbalanced, with relatively few cases lacking metastasis at diagnosis. While this limitation is understandable, the authors should refrain from making overly bold claims based on the small sample size.

We agree, and have tried to emphasize appropriate limitations. Notably, key observations from scRNA-seq are supported by orthogonal data from other approaches. Upregulation of fetal pathway signatures was originally from RNA-seq, and ASCL1 involvement is supported by ChiP-seq, western blotting, and FFPE-ATAC.

- The identification of epithelial cells as tumor cells requires additional validation. InferCNV alone is insufficient to confirm tumor identity. The authors should use complementary approaches, such as FISH analysis for known chromosomal aberrations, to confirm tumor cell populations.
- Potential confounding by healthy epithelial cells. Based on publicly available single-cell atlases of healthy lungs (e.g., linked datasets below), ACSL1 and FOXA1 are highly co-expressed in alveolar type II epithelial cells. This raises the possibility that observed differences may reflect variations in the proportion of cancer versus healthy epithelial cells. The authors should account for this by explicitly quantifying cancer epithelial cell contributions.

There are no common chromosomal aberrations in SCLC suitable for detection by FISH. InferCNV is an established and widely cited method for differentiating cancer cells and normal cells in the field. There are multiple papers supporting this method as a primary tool (Anoop P et al., Science, 2014, Tirosh I, et al, Science,

2016, Puram SV, et al. Cell, 2017), and further information at Home · broadinstitute/infercnv Wiki · GitHub. Our team has previously used InferCNV to differentiate cancer and normal cells in our first paper on scRNA-seq of human SCLC (Chan J et al., Cancer Cell, 2021). As shown in the figures below, at left all cells, and right epithelial cells only, non-malignant cells with relatively homogenous copy number across the genome (top) can be clearly distinguished from the heterogeneity of the SCLC genome, demonstrating extensive and complex copy number aberrations among subclones. This method clearly distinguishes genomically unstable cancer cells from potentially contaminating healthy epithelial cells, such as AT2 cells.

2. PCA Analysis (Extended Fig. 2D)

Principal component analysis (PCA) suggests the presence of technical or other confounding factors. The authors should:

- Provide additional PCA plots (e.g., PC2/PC3, PC3/PC4) to better evaluate batch effects or technical artifacts.
- Include a heatmap showing primary non-metastatic, primary metastatic, and metastasis samples in the same plot for better assessment of the variability.

Below we provide the additional PCA plots as suggested. These further support the shown PCA. The scree plot at left shows the explained variance of each PC. At PC4 to PC5, the explained variance starts to level off. PC1 to PC4 can confidently differentiate never-met and metastasis groups.

We also include the heatmap as suggested. This similarly shows separation of never-mets and metastasis groups. We have included these data in Extended data Fig2.

3. Data Visualization

- Fig. 2E: Replace the current visualization with violin plots, showing each sample as a distinct data point, to improve interpretability.

We appreciate this suggestion, and have replaced the heatmap with a violin plot. This plot more clearly shows FOXA2 as the only candidate gene from the epithelium.

- Fig. 4G-J: Clarify whether "set size" refers to the number of differentially expressed (DE) genes in a pathway or the total number of genes in the pathway database. Ideally, display the number of DE genes per pathway to avoid misinterpretation.

"Set size" refers to the total number of genes in the pathway. We have included this in the figure legend. The exact number and genes can also be traced in the supplementary table.

- Fig. 5B: The integration of single-cell RNA-seq data appears flawed or incomplete. Without code or a detailed methods description, it is difficult to assess this issue.

We uploaded the code as suggested. We also show the integration of single-cell RNA-seq data below. In the first panel, we embedded the data without any batch correction for these 19 SCLC patients.

(False; FOXA2-, True; FOXA2+)

Next, we analyzed the putative malignant cells identified using the InferCNV method discussed above. Then we repeated the correction with Harmony and increased the iterations to allow convergence.

[REDACTED]

We also used another method for batch correction called scVI and reached to the same result which shows that FOXA2+ and FOXA2- population has a different gene expression profile.

[REDACTED]

We substituted the UMAP from Fig5.

4. Experimental Design and Comparisons

- *The comparison of "never-met primary vs. met-associated metastasis" is unconventional. A more logical design would compare "never-met primary vs. met-associated primary" and "met-associated primary vs. met-associated metastasis" (ideally matched samples). The authors should justify their chosen comparisons or revise their analysis.*

We agree with the reviewer: the most informative comparison is "never-met primary vs. met-associated primary," which is the initial comparison used to define candidate competency factors (blue circles in Fig 2f). One might expect that factors associated with metastatic competency would be expressed not only in "met-associated primaries" but also in metastases: thus "never-met primary vs. met-associated metastasis" serves as a second comparison (red circles in Fig 2f) in a Venn diagram to narrow down lead candidates. We would argue that "met-associated primary vs. met-associated metastasis" might be less relevant, since both of these harbor metastatic competency: the primary in these cases having in fact given rise to metastases. We tried to clarify the underlying hypothesis regarding metastatic competency (and the lack thereof specifically in never-met primaries) in the revised manuscript, in introducing the heat maps used in Fig2c-e:

Line 196; We hypothesized that primary SCLC tumors that never metastasized might lack one or more factors involved in metastatic competency, which would be present in primary SCLC tumors that gave rise to metastatic disease, and which would presumably also be retained in metastatic SCLC. In addition to comparing gene expression in never-met primary tumors vs. met-associated primary tumors and metastases considered jointly (**Fig.2c**). To minimize the impact of distinct microenvironments on the transcriptional profiles, we initially analyzed differential gene expression between the never-met primary SCLC tumors versus met-associated primary SCLC tumors (**Fig.2d, Extended Data Fig.2f**).

5. Other Methodological Concerns

- *GSEA Analysis: Genes are typically ranked by logFC rather than p-values. The authors should clarify their rationale for using p-values and specify whether pre-filtering based on logFC was performed, prior to gene ranking.*

We did not pre-filter DEGs for GSEA, and would argue that it is appropriate to rank by p-value. Ranking genes based on the log₁₀-transformed p-values and their direction offers several advantages as outlined in (Zyla J et al. BMC Bioinformatics, 2017, PMID: 28499413) including:

1. **Emphasis on Statistical Significance** This method highlights genes that are not only differentially expressed but also statistically significant. By using the log₁₀(p-values), you give more weight to genes with lower p-values, which are more likely to be true positives.

2. **Retention of Directionality** By multiplying the $\log_{10}(\text{p-values})$ by the sign of the \log_2 fold changes, you retain the direction of the gene expression changes (upregulated or downregulated). This helps in understanding the biological context of the changes.

3. **Improved Sensitivity and Specificity** Combining the magnitude of change (\log_2 fold change) with the statistical significance (p-value) can improve the sensitivity and specificity of the ranking. This means you are more likely to identify biologically relevant gene sets.

4. **Enhanced Biological Insights** This ranking method can lead to more meaningful biological insights, as it considers both the extent of differential expression and the confidence in those changes. This dual consideration can help in identifying key pathways and processes affected.

5. **Robustness to Noise:** By focusing on statistically significant changes, this method can be more robust to noise in the data. Genes with high variability but low significance are less likely to dominate the results.

- *ChIP-seq/qPCR: Input DNA amounts are highly variable (0.02–5 ng), which can introduce bias. For cell line experiments, such low input values are unjustifiable. The authors should provide additional justification or repeat these experiments with consistent, sufficient input amounts.*

Thank you for the comment; this was an error. We confirmed the amounts used for the ChIP-seq analysis as follows. We have corrected the input DNA amounts as ranging from 0.2-5ng.

[REDACTED]

Fig. 6C, E: The data quality is unclear. The authors should include positive control regions (constitutively open/accessibly chromatin) to demonstrate data reliability. Add scale bars to plots for accurate interpretation of chromatin accessibility.

Thank you for this suggestion. We now present the ATAC-seq data including scale bars at *TUBA1A* and *ACTB* loci which show nuclease-accessible chromatin in both genes. We have included in the text:

Line 410; This method showed accessibility to the internal control genes (*TUBA1A* and *ACTB*) supporting the quality of this assay (Extended Data Fig.6c).

Line 423; Supporting the primary tumor analysis, we found that the *FOXA2* locus was open in all 3 ASCL1⁺/FOXA2⁺ PDX tumors, but closed in the ASCL1⁺/FOXA2⁻ PDX tumor (Fig.6e, Extended Data Fig.6g).

6. Prospective Implications

- Could metastasis be predicted in a prospective study based on ASCL1 and FOXA2 expression at diagnosis?

Thank you for the suggestion. We have added this potential implication to the manuscript:

Line 509; Our data support FOXA2 as a candidate predictor of SCLC relapse in patients with definitively treated early stage disease.

- Would longer follow-up alter conclusions regarding "never-met" patients with slower-growing tumors? These points should be discussed to contextualize the findings.

If this hypothesis were true, "never-mets" should be substantially less proliferative, which can be assessed by Ki67 index on IHC. SCLC is notoriously never slow growing – the majority of cells are always actively in cell cycle as reflected by Ki67 indices over 50%. There was no significant difference in Ki67 index between never-met and metastatic cases. We required a minimum of 2 years of followup at MSK to consider a case to be "never metastasizing", and several patients had much longer followup. The three never-met cases with lowest Ki67 were followed up for 790 days, 1287 days and 2236 days, respectively. Based on these data, we do not think that indolent cancer is the cause of never-mets. We have included these Ki67 data as Extended Data Fig. 2b and added a comment in the manuscript:

Line 179; There was no significant difference in Ki-67 index between never-met and met-associated tumors (Extended Data Fig.2b).

Reviewer #4

(1) The mechanistic details connecting FOXA2 to the observed fetal neuroendocrine gene expression program and subsequent metastasis remain relatively superficial. The connection between FOXA2, the fetal neuroendocrine gene expression program, and the specific processes involved in metastasis (e.g., EMT, cell migration, invasion, extravasation) remains unclear. The current evidence is primarily correlative. Further investigation into the downstream pathways activated by FOXA2 and its interaction with other metastasis-related genes is needed.

We would note that the data supporting FOXA2 as a driver of metastasis is not just correlative – we have shown

by gene knock-down and restoration that FOXA2, while having no apparent effect on proliferation, has a strong effect on metastasis *in vivo*. This is a novel and fundamental insight. We agree the further interrogation of the several possible downstream effects of FOXA2 is warranted: many of the downstream processes indicated by the reviewer may be contributing, and assessing their relative contributions will be of substantial interest as the focus of future studies.

We do think that the fetal neuroendocrine gene expression signature consistently induced by FOXA2 is of particular relevance. Our prior work across multiple cancer types has consistently implicated early progenitor pathways in metastasis initiation (summarized in Massagué and Ganesh, *Cancer Discov* 2021; of relevance to lung cancer see Laughney et al., *Nat Med* 2020). This signature is comprised of >20 genes; individual interrogation of all members of this network is not readily feasible; in light of prior data we consider it a key observation that FOXA2 itself activates this precursor program. We have tried to emphasize the relevance of this in the revised manuscript, as follows;

Line 457; Fetal gene signatures driving developmental pathways in tissue differentiation have been extensively implicated as key to establishment and expansion of metastases, across multiple tumor types (reviewed in [Massagué and Ganesh, *Cancer Discov*, 2021]). Together these studies support that aberrant reactivation of early precursor pathways supporting normal tissue development enable metastasis precursors to survive, adapt, and proliferate in diverse microenvironments.

(2) Although *in vivo* metastasis assays are performed, the manuscript lacks further functional assays to directly test the impact of FOXA2 on specific aspects of metastasis (e.g., cell migration, invasion, adhesion). Such assays would strengthen the mechanistic understanding.

FOXA2 is a pioneer transcription factor known to impart a progenitor state in various cell lineages [Ang SL, *Cell*, 1994, Weinstein DC, *Cell* 1994, Laughney, *Nat Med*, 2020], and progenitor phenotype (i.e. “stemness”) is a fundamental requirement for metastatic colony formation, which is then complemented by the separate action of mediators of cancer cell invasion, extravasation, and host microenvironment adaptation. To provide an *in vitro* correlate of the *in vivo* metastasis assays, we include colony formation assays using cell lines with FOXA2 either knocked down or overexpressed. Results were consistent with the *in vivo* observations. We included these results in the revised manuscript as follows;

Line 342; In line with the *in vivo* data, colony formation assays showed that FOXA2 KD in both K1836 and SHP-77 resulted in fewer colonies and FOXA2 OE in H1963 leads to increased number of colonies *in vitro* (**Extended Data Fig.4d**).

(3) The manuscript identifies ASCL1 as a regulator of FOXA2, but the precise molecular mechanisms through which ASCL1 activates FOXA2 remain unclear. Are all ASCL1 high tumors have high FOXA2 levels? Are there ASCL1 high tumors that show low expression of FOXA2? Does FOXA2 have different effects in different cell populations within the tumor? Are there specific enhancer regions involved in FOXA2 regulation? What other transcription factors or epigenetic modifications are at play? The study lacks the depth of mechanistic analysis needed to truly understand the regulatory network surrounding FOXA2.

As noted above in response to **Reviewer 1 Points 1 and 2**, we have tried to further clarify the relationship between ASCL1 and FOXA2 in the revised manuscript. ASCL1 drives a broad program of gene expression

implicated in SCLC tumorigenesis, cell survival, and proliferation, with downstream effectors including FOXA2, PROX1 and others. We show here that the specific activation of FOXA2 by ASCL1 can promote metastasis, without effects on cancer cell viability or proliferation. Not all ASCL1-expressing SCLC are positive for FOXA2, implying that other positive or negative regulators of ASCL1-dependent FOXA2 expression must act in concert with ASCL1 to influence FOXA2 expression. Interrogation of these additional transcriptional regulatory factors might be best approached with an initial finely targeted mutational screen of the FOXA2 promoter/enhancer elements, which will be of substantial interest for future studies.

(4) The majority of the in vivo experiments utilize only two SCLC cell lines (H1836 and SHP-77). In addition, SHP77 cell line expresses untypical for SCLC oncogenic KRAS(G12V) mutant and lacks of characteristic for SCLC RB1 mutations. It is crucial to replicate the findings in additional cell lines and, ideally, in a wider range of PDXs to confirm the generalizability of the findings beyond this limited set. In addition, ASCL1-low and ASCL1-high cell lines should be used to validate the proposed ASCL1-FOXA2 signaling axis.

To address the reviewer's comment, we tried with 3 additional FOXA2-high cell lines (including a PDX-derived cell line; H146, H2081, Lx674c) to generate FOXA2 KD. We performed drug selection, and FACS sorting only for GFP-high cells, however, this approach did not yield sufficient knock-down to move on to in vivo experiments. It is possible that these cell lines do not tolerate well a deep depletion of FOXA2.

[REDACTED]

As the reviewer suggested, we also used the FOXA2-low line H1963 for intra-cardiac injection. As described in response to **Reviewer 1 Point 3**, this line generated relatively few metastases (1 of 6 mice and 1 of 8 mice, in repeat experiments, with both metastases being to the ovary). Interestingly, both metastatic lesions demonstrated upregulated FOXA2 expression, suggesting that FOXA2 may have facilitated establishment of metastasis. However, exogenous overexpression of FOXA2 in H1963 did not increase the frequency of metastases observed, suggesting that in FOXA2-low SCLC, other factors may influence metastatic capacity. We do not claim that FOXA2 is the only factor that affects metastatic potential in SCLC – rather that it is one such factor, that might be of particular relevance in the context of ASCL1-expressing SCLC. In sum, we identify FOXA2 as necessary but not sufficient for SCLC metastasis.

[REDACTED]

(5) The authors acknowledge that the never-metastatic cohort might not represent truly non-metastatic disease. Some tumors in this group might represent early-stage disease caught before widespread metastasis. This ambiguity significantly weakens the comparison and raises concerns about the generalizability of the findings. A more rigorous definition, perhaps incorporating tumor stage and survival time beyond a simple 2-year threshold, along with additional biological characterization of the "never-metastatic" tumors is necessary.

The “never-metastatic” cohort was carefully delineated here, to the extent possible, to exclude any cases that

might eventually metastasize. These cases demonstrated no relapse within a minimum documented followup of 2 years; this 2-year milestone was chosen as a minimum given that 90-95% of SCLC cases recur and progress within this timeframe. Some were followed for up to 6 years, and none ever developed recurrent disease. To the reviewer's point, stages were in fact limited to T1-3 N0 M0, i.e. tumors could become quite large, but none demonstrated even local (intraobar) lymph node spread. Overall survival beyond the 2-year minimum threshold, in the absence of metastasis (which we continued to track in all patients beyond the 2-year minimum) would not add to the definition of never-metastatic, as it could only further define patients who died of something other than recurrent (metastatic) disease. Overall survival data is included, as a point of interest, in Extended Data Fig2c; only one patient in the never-met cohort has died in followup, 7 years after diagnosis. Since this was the basis of our initial screen for candidates, we in fact endeavored to be as rigorous as possible in defining this cohort. We underscored these criteria in the revised manuscript, for clarity:

Line 170; We identified 8 SCLC cases meeting **strict criteria as** never-metastatic, including pathologic stage T1-3N0M0 (**no evidence of spread, even to local intraobar nodes**), definitive treatment with surgical resection or concomitant chemoradiation, no relapse with a minimum of 2 years of documented follow-up, and available tumor material.

In closing we wish to again thank the reviewers for their input and suggestions. These have substantially improved the resulting manuscript.

RESPONSE TO REVIEWERS

We would like to thank the reviewers for their several thoughtful suggestions regarding this work. We are pleased that 3 of the reviewers recommend publication without reservations. Reviewer comments are reproduced verbatim, followed by responses in blue.

Reviewer #1

I would like to acknowledge the high-quality work performed by the authors in addressing my comments and complementary comments from 3 other reviewers. I have no further concerns.

We appreciate the reviewer's support.

Reviewer #2

The authors have carefully and thoughtfully addressed all my comments and concerns. These revisions and clarifications have improved the manuscript in my opinion. I have no further concerns.

We appreciate the reviewer's support.

Reviewer #3

I appreciate the authors thorough revision, which have adequately addressed all my concerns and questions. I have no further comments and recommend the manuscript for publication in Nature Communications.

We appreciate the reviewer's support and recommendation for publication.

Reviewer #4

-It is unclear why FOXA2 depletion was not technically possible in any additional cell lines, given that the Authors claim FOXA2 does not affect normal cell growth. Therefore, it is likely that FOXA2 is a critical survival factor in those cells, undermining the Author's claim that FOXA2 is predominantly a regulator of metastasis.

We have shown by gene knock-down and restoration that FOXA2 has no apparent effect on cell proliferation while it has a strong effect on metastasis *in vivo*. We demonstrated this in two SCLC cell lines, H1836 and SHP-77, and using two different hairpins, which is quite reasonable. We did try to establish FOXA2 knockdown in a third cell line, but the reduction in FOXA2 expression achieved was not sufficient to move to *in vivo* experiments. We believe that successful knockdown in two independent lines, with two hairpins, resulting in consistent effects, is sufficient to support the point.

[REDACTED]

-In addition, the Authors write (in response to R2's important question regarding the mechanisms of ASCL1 binding to FOXA2) that: single-cell sorting after CRISPR targeting is not possible in H1836 cells. That further supports the need to perform the experiments in additional cell lines in which mechanistic experiments will be possible.

In our reply to Reviewer 3 (who was the one to raise this point), we explained that, in our experience, SCLC cell suspensions grow in clusters and do not recover from single-cell sorting, which makes it difficult to perform targeted editing of FOXA2 regulatory elements. While this experiment would have extended our work, its absence does not detract from the validity of our central claim that FOXA2 promotes SCLC metastasis and differences in ASCL1 access to the FOXA2 locus are associated with FOXA2 expression. Indeed, responding to our explanation Reviewer 3 writes: "I appreciate the authors thorough revision, which have adequately addressed all my concerns and questions. I have no further comments and recommend the manuscript for publication in Nature Communications."

-Requested direct testing of the impact of FOXA2 on specific aspects of metastasis like cell migration or invasion assays, was not performed. Instead, the Authors performed colony formation assays, which showed that FOXA2 depletion robustly (up to~ 90%) inhibits the tumor-forming properties of SCLC cell lines. That is in contradiction with the xenograft assays, which indicate that there is no defect in tumor-forming properties.

We appreciate the Reviewer's view that cell migration and invasion are important for metastasis and should be studied here. However, we wish to respectfully raise several arguments against this position.

First, *in vitro* migration and invasion assays are at best a poor surrogate for assessment of metastatic competency. We present a much more powerful and definitive result - our *in vivo* data showing a strong decrease in metastatic colonization in FOXA2-depleted SCLC cells are obtained by intra-cardiac injection of the test cells directly into the blood stream. This method focuses on metastasis colonization steps which crucially depend on stemness and niche adaptation capacity of the disseminated cancer cells.

Second, given what is known about FOXA2 in developmental biology, even if cell migration in a dish or Boyden chamber were observed as a function of FOXA2 expression *in vitro*, it would be problematic to conclude that FOXA2 promotes metastasis by regulating cell migration. FOXA2 is not part of the cell migration and invasion apparatus. It is a transcription factor that primarily regulates progenitor cell phenotypes. Performing such assays with floating non-adherent cells is itself inherently problematic. Even if an effect were seen (unlikely in cells that do not adhere), we would not conclude that FOXA2 promotes metastasis by primarily driving invasion or migration.

Third, our observed phenotypes are supported by what is known about FOXA2 in other systems. FOXA2 likely enables metastasis by endowing SCLC cells with multiple relevant traits including "stemness" (capacity for clonogenic growth), phenotypic plasticity, niche adaptability, microenvironment signal responsiveness, etc. Clonogenic activity is known to be essential for metastatic colony outgrowth from disseminated cancer cells although it is not required for tumorigenic growth of samples consisting of thousands of cancer cells as is the case with xenograft and other sub-cutaneous tumor implantation assays.

Based on these reasons, we chose colony formation assay as a relevant *in vitro* surrogate for metastatic colony formation.

-Of note, further description in the Methods section of colony formation assays would be necessary as it is not obvious how this experiment was performed with suspension cells.

We thank the reviewer for catching this omission. We added the description of colony formation assay in detail as requested.

-No further mechanistic insight on how FOXA2 regulates target genes is presented in the revision. Chromatin accessibility changes are insufficient to directly establish causation. Given that the previous publications (PMID: 37963187) established the critical role of FOXA1/2 in SCLC metastasis, the presented manuscript is somewhat more incremental.

We agree that the present work now provides a framework for investigating the downstream targets of FOXA2, which of these downstream factors are required for metastasis, how these factors promote metastasis, and whether these factors, more than their upstream transcriptional drivers, might make good targets for therapy. We agree that this would be logical extensions of this work and appropriate for future studies. However, we consider such an extension to be beyond the scope of the present manuscript.

The differences between our work and the PMID:37963187 paper were already appreciated by Reviewer 1, who raised this point in the initial round of reviews. Reviewer 1 now states: "*I would like to acknowledge the high-quality work performed by the authors in addressing my comments and complementary comments from 3 other reviewers. I have no further concerns.*" We wish to re-emphasize the novelty of our study compared to the Ko et al. (PMID:37963187) paper by Julien Sage and colleagues:

1. Our data are based on human SCLC systems, including unique clinical samples and datasets, whereas PMID:37963187 exclusively used mouse models. While highly valuable, mouse models of SCLC have critical differences with human disease, including low tumor mutation burden, relative genomic stability, rarity of brain metastases, and absence of tobacco mutational signature.
2. The main focus of PMID:37963187 was on NFIB and a distinction between NFIB-driven and NFIB-independent pathways. The insights on FOXA1/2 in that paper are minimal and limited to suggesting that FOXA1/2 could be relevant to metastasis and worth investigating. The actual demonstration of FOXA2 as a crucial mediator of SCLC metastasis is provided for the first time in our manuscript. We identified FOXA2 as a key candidate based on an unbiased analysis of a unique collection of metastatic and never-metastatic clinical samples and datasets that we collected.

RESPONSE TO REVIEWERS

Reviewer #4

I appreciate the authors' candid and honest response. I fully acknowledge that SCLC lines can be temperamental during viral transductions and in the selection process. I kindly suggest that the authors include a brief overview of the negative results observed in knockdown experiments conducted in other cell lines. I believe the editor will concur that presenting these results will not detract from the manuscript; rather, it will inform readers that attempts were made to reproduce findings in suspension SCLC lines, which are more commonly used for SCLC growth studies. Including this information would enhance the manuscript by providing a more comprehensive understanding of the challenges associated with these experimental approaches in SCLC.

Thank you very much for the thoughtful comment. We have now included the following sentence as suggested.

Line 286; Attempts to establish FOXA2 knockdown in a 3rd cell line (H2081, H146, and one PDX derived line) did not yield a sufficient reduction in FOXA2 expression to move to *in vivo* experiments.